# Djakonoviosides A, A_1_, A_2_, B_1_–B_4_ — Triterpene Monosulfated Tetra- and Pentaosides from the Sea Cucumber *Cucumaria djakonovi*: The First Finding of a Hemiketal Fragment in the Aglycones; Activity against Human Breast Cancer Cell Lines

**DOI:** 10.3390/ijms241311128

**Published:** 2023-07-05

**Authors:** Alexandra S. Silchenko, Anatoly I. Kalinovsky, Sergey A. Avilov, Roman S. Popov, Pavel S. Dmitrenok, Ekaterina A. Chingizova, Ekaterina S. Menchinskaya, Elena G. Panina, Vadim G. Stepanov, Vladimir I. Kalinin, Valentin A. Stonik

**Affiliations:** 1G.B. Elyakov Pacific Institute of Bioorganic Chemistry, Far Eastern Branch of the Russian Academy of Sciences, Pr. 100-letya Vladivostoka 159, 690022 Vladivostok, Russia; kaaniv@piboc.dvo.ru (A.I.K.); avilov_sa@piboc.dvo.ru (S.A.A.); popov_rs@piboc.dvo.ru (R.S.P.); paveldmt@piboc.dvo.ru (P.S.D.); chingizova_ea@piboc.dvo.ru (E.A.C.); ekaterinamenchinskaya@gmail.com (E.S.M.); stonik@piboc.dvo.ru (V.A.S.); 2Kamchatka Branch of Pacific Institute of Geography, Far Eastern Branch of the Russian Academy of Sciences, Partizanskaya st. 6, 683000 Petropavlovsk-Kamchatsky, Russia; egpanina777@gmail.com (E.G.P.); stepanovvadim24@gmail.com (V.G.S.)

**Keywords:** *Cucumaria djakonovi*, Dendrochirotida, triterpene glycosides, djakonoviosides, sea cucumber, hemolytic and cytotoxic activity, human breast cancer

## Abstract

Seven new monosulfated triterpene glycosides, djakonoviosides A (**1**), A_1_ (**2**), A_2_ (**3**), and B_1_–B_4_ (**4**–**7**), along with three known glycosides found earlier in the other *Cucumaria* species, namely okhotoside A_1_-1, cucumarioside A_0_-1, and frondoside D, have been isolated from the far eastern sea cucumber *Cucumaria djakonovi* (Cucumariidae, Dendrochirotida). The structures were established on the basis of extensive analysis of 1D and 2D NMR spectra and confirmed by HR-ESI-MS data. The compounds of groups A and B differ from each other in their carbohydrate chains, namely monosulfated tetrasaccharide chains are inherent to group A and pentasaccharide chains with one sulfate group, branched by C-2 Qui2, are characteristic of group B. The aglycones of djakonoviosides A_2_ (**3**), B_2_ (**5**), and B_4_ (**7**) are characterized by a unique structural feature, a 23,16-hemiketal fragment found first in the sea cucumbers’ glycosides. The biosynthetic pathway of its formation is discussed. The set of aglycones of *C. djakonovi* glycosides was species specific because of the presence of new aglycones. At the same time, the finding in *C. djakonovi* of the known glycosides isolated earlier from the other species of *Cucumaria*, as well as the set of carbohydrate chains characteristic of the glycosides of all investigated representatives of the genus *Cucumaria*, demonstrated the significance of these glycosides as chemotaxonomic markers. The membranolytic actions of compounds **1**–**7** and known glycosides okhotoside A_1_-1, cucumarioside A_0_-1, and frondoside D, isolated from *C. djakonovi* against human cell lines, including erythrocytes and breast cancer cells (MCF-7, T-47D, and triple negative MDA-MB-231), as well as leukemia HL-60 and the embryonic kidney HEK-293 cell line, have been studied. Okhotoside A_1_-1 was the most active compound from the series because of the presence of a tetrasaccharide linear chain and holostane aglycone with a 7(8)-double bond and 16*β*-*O*-acetoxy group, cucumarioside A_0_-1, having the same aglycone, was slightly less active because of the presence of branching xylose residue at C-2 Qui2. Generally, the activity of the djakonoviosides of group A was higher than that of the djakonoviosides of group B containing the same aglycones, indicating the significance of a linear chain containing four monosaccharide residues for the demonstration of membranolytic action by the glycosides. All the compounds containing hemiketal fragments, djakonovioside A_2_ (**3**), B_2_ (**5**), and B_4_ (**7**), were almost inactive. The most aggressive triple-negative MDA-MB-231 breast cancer cell line was the most sensitive to the glycosides action when compared with the other cancer cells. Okhotoside A_1_-1 and cucumarioside A_0_-1 demonstrated promising effects against MDA-MB-231 cells, significantly inhibiting the migration, as well as the formation and growth, of colonies.

## 1. Introduction

Marine invertebrates belonging to the class Holothuroidea are named the “pearls of the sea” because some of them present valuable sea food and almost all, if not all, produce triterpene glycosides, demonstrating various biologic activities. Despite the fact that the number of sea cucumber species whose glycosidic compositions have been studied is steadily growing, the majority of them are still unexplored because of their inaccessibility or the complexity of isolating the individual compounds from the extracts obtained from the producer organisms. Every new investigation of the glycosidic composition of unstudied species of sea cucumbers or the reinvestigation of the glycosidic composition of the species studied earlier, using the modern methods of isolation of individual substances from the multicomponent mixtures, resulted in the finding of dozens of new structural variants of the glycosides, which have unique features [1,2,3,4,5]. The early studies of some representatives of the genus *Cucumaria* showed the species specificity of glycosidic composition that allowed the use of these compounds for resolving the taxonomic challenges arising because of the high phenotypical polymorphism within one species of *Cucumaria*, probably because of a long evolutionary history [6]. The comparing of the glycosides from some *Cucumaria* species revealed they are characterized by significant structural variability of the aglycones [6,7,8,9,10,11,12,13,14], while they shared the same carbohydrate chain structures, including predominant mono-, di-, and trisulfated pentaosides with the branching xylose unit attached to the second monosaccaride (Qui). Tetrasaccharide monosulfated sugar chains are also present in the glycosides of several *Cucumaria* species [11,15], but these compounds are not so abundant, as a rule.

It is known that the glycosides from *Cucumaria* possess cytotoxic, proapoptotic, and immunomodulatory properties. The immunomodulatory preparation Cumaside, created on the basis of cucumarioside A_2_-2 isolated from the sea cucumber *Cucumaria japonica*, also demonstrating the anticancer action [16]. Nowadays, triterpene glycosides from sea cucumbers attract the attention of scientists worldwide, being potential antitumor agents demonstrating cytotoxic and antiproliferative action against different human cancer cells in vitro, initiating apoptosis, inhibiting tubules formation, adhesion, migration, invasion, and the angiogenesis of cancer cells [17,18,19,20,21,22,23]. Breast cancer (BC) is the mostly widespread (more than 2.2 million of cases were registered in 2020) and leading cause of death from oncologic diseases for women. The treatment for breast cancer is personalized because it depends on the disease’s stage and the molecular-biological type of the cancer, which is determined by the presence in tumor cells of the receptors for estrogen, progesterone, or human epidermal growth factor (HER2) [24]. The described biochemical features of BC give an advantage for the development of a target therapy against each molecular type of BC. Since searching for suppressors of breast cancer cells is an important scientific and medical task, the investigation of cytotoxic action against human breast cancer cell lines MCF-7, T-47D, and triple-negative MDA-MB-231 was undertaken.

The sea cucumber *C. djakonovi* has limited geographical distribution from the Bering Sea to Avacha Gulf on the eastern coast of Kamchatka Peninsula. This species has been often erroneously identified as *C. japonica* or *C. frondosa* when classical taxonomic features were analyzed. Hence, the investigation of the composition of triterpene glycosides of *C. djakonovi* will be very helpful for resolving taxonomic difficulties. Herein, the isolation, structure elucidation, and biologic activity testing of new monosulfated tetra- and pentaosides djakonoviosides A (**1**), A_1_ (**2**), A_2_ (**3**), B_1_ (**4**), B_2_ (**5**), B_3_ (**6**), and B_4_ (**7**) are reported. The chemical structures of **1**–**7** were elucidated by the analyses of the ^1^H, ^13^C NMR, 1D TOCSY, and 2D NMR (^1^H,^1^H COSY, HMBC, HSQC, ROESY) spectra, as well as the HR-ESI mass spectra. All the original spectra are displayed in Appendix A. The hemolytic activity against human erythrocytes and cytotoxic activities against human cell lines—breast cancer cells MCF-7, T-47D, and triple-negative MDA-MB-231, as well as leukemia HL-60 and embryonic kidney HEK-293 cells—were tested.

## 2. Results and Discussion

### 2.1. Structure Elucidation of the Glycosides

The crude glycosidic sum of *Cucumaria djakonovi* (13.79 g) was obtained after the hydrophobic chromatography of the concentrated ethanolic extract on a Polychrom-1 column (powdered Teflon, Biolar, Latvia). Its initial separation was achieved using the chromatography on Si gel columns (CC) with the stepped gradient of the eluents system CHCl3/EtOH/H2O in ratios 100:50:4, 100:75:10, 100:100:17, and 100:125:25 resulting in the obtaining of five fractions. The fractions I and II were repeatedly subjected to CC with the system of eluents CHCl3/EtOH/H2O (100:75:10) that resulted in getting subfractions 1 (244 mg) and 2 (640 mg). The individual glycosides **1**–**7** (Figure 1), along with three known compounds, have been isolated after HPLC of these subfractions on reversed-phase semipreparative columns Supelco Ascentis RP-Amide (10 × 250 mm) and Phenomenex Synergi Fusion RP (10 × 250 mm).

Three known triterpene glycosides were found in the glycosidic fraction of *C. djakonovi*. The structures of the known compounds were examined by the analysis of NMR and MS data followed by comparison with the literature data that led to identification of okhotoside A_1_-1 isolated first from *C. okhotensis* [10], cucumarioside A_0_-1 from *C. japonica* [25], and frondoside D from *C. frondosa* [26] (Appendix A).

The sugar configurations in glycosides **1**–**7** were assigned as *D* on the basis of the analogy with all other known triterpene glycosides from sea cucumber.

Extensive analysis of the ^1^H,^1^H COSY, 1D TOCSY, HSQC, and ROESY spectra of the carbohydrate moieties of compounds **1**–**3** (Table 1, Appendix A) indicated that the identical monosulfated tetrasaccharide chains are characteristic of these glycosides. The monosaccharide composition was determined as two xylose (Xyl1 and Xyl3), one quinovose (Qui2), and 3-*O*-methylglucose (MeGlc4). The positions of the glycosidic linkages established by the ROESY and HMBC correlations corresponded to the linear tetrasaccharide chain with *β*-glycosidic bonds: Xyl1 bonded to C-3 of the aglycone, Qui2 linked to C-2 Xyl1, Xyl3—to C-4 Qui2, and terminal MeGlc4—to C-3 Xyl3 (Table 1). The presence of a sulfate group was established by distinctive values of δ_C_ of C-4 Xyl1 observed at δ_C_ 75.9 and C-5 Xyl1 observed at δ_C_ 64.1 (*α*- and *β*-shifting effects of sulfate group) and corroborated by the MS data of each glycoside. Such a structure of sugar moiety is common for the glycosides from the sea cucumbers of different taxa. The glycosides, bearing this carbohydrate chain and isolated from *C. djakonovi*, were named djakonoviosides of group A.

The molecular formula of djakonovioside A (**1**) was determined to be C_55_H_87_O_26_SNa from the [M_Na_–Na]^−^ ion peak at *m*/*z* 1195.5209 (calc. 1195.5212) in the (−)HR-ESI-MS (Appendix A). The spectra of the aglycone part of **1** (Table 2, Appendix A) demonstrated the signals of 18(20)-lactone at δ_C_ 179.7 (C-18) and δ_C_ 85.3 (C-20), 7(8)-double bond at δ_C_ 120.2 (C-7) and δ_C_ 145.7 (C-8), and *O*-acetyl group δ_C_ 169.5 (O**C**OCH_3_) and δ_C_ 21.3 (OCO**C**H_3_), attached to C-16 (δ_C_ 75.2) in the polycyclic nucleus. The position of this functionality was corroborated by the cross-peaks between H-15 and C-16 and the methyl group of acetoxy substituent and C-16 in the HMBC spectrum of **1**. Common for the glycosides, the *β*-orientation of the 16-acetoxy group in djakonovioside A (**1**) was confirmed by ROE correlation H-16/H-32. The protons of the side chain (H-22–H-27) formed an isolated spin system in the COSY spectrum, showing that the signal of H-23 was shifted downfield to δ_H_ 4.08. A corresponding carbon signal deduced by the HSQC spectrum was observed at δ_C_ 65.9, indicating the presence of a hydroxyl group at this position. The known frondoside D [26], isolated from *C. djakonovi* along with new compounds, has the same aglycone as djakonovioside A (**1**). The (23S) configuration was suggested for frondoside D based on the comparison of the δ_C_ of C-22–C-24 with those for stichlorogenol—the aglycone of the glycosides from *Stichopus chloronotus*—the stereochemistry of which had been established by X-ray crystallography [27]. To assign the configuration of the C-23 chiral center in compound **1**, the modified Mosher’s method was applied [28]. The treatment of **1** with (R)- and (S)-MTPA chloride gave the 23-S-(−)- and 23-R-(+)-MTPA esters, respectively. The chemical shifts of the protons surrounding chiral center C-23 were found by the ^1^H,^1^H COSY spectra of the MTPA esters. The positive values of Δ*^SR^* for H-15–H-22 and negative values of Δ*^SR^* for H-24–H-27 indicated the 23*S* configuration in **1** to be the same as in frondoside D (Appendix A).

The (−)ESI-MS/MS of **1** (Appendix A) demonstrated the fragmentation of the [M_Na_–Na]^−^ ion with *m*/*z* 1195.5 giving the fragment ion peaks at *m*/*z* 1049.4 [M_Na_–Na–C_5_H_11_O–CH_3_COO]^−^ corresponding to the cleavage of a side chain by a C-22/C-23 covalent bond and the loss of the *O*-acetyl group, 1033 [M_Na_–Na−CH_3_COO–SO_3_Na]^−^, and 989 [M_Na_–Na––CH_3_COO–SO_3_Na–CO_2_]^−^, demonstrating the elimination of the CO_2_ molecule related to the cleavage of the lactone cycle bonds along with the loss of the *O*-acetyl and sulfo-groups, 975.4 [M_Na_–Na–CO_2_–MeGlc]^−^, 741.3 [M_Na_–Na–MeGlc–Xyl–Qui]^−^, and 681.2 [M_Na_–Na–Agl–H]^−^, corroborating the whole structure of djakonovioside A (**1**).

These data indicate that djakonovioside A (**1**) is 3*β*-*O*-[3-*O*-methyl-*β*-D-glucopyranosyl-(1→3)-*β*-D-xylopyranosyl-(1→4)-*β*-D-quinovopyranosyl-(1→2)-4-*O*-sodium sulfate-*β*-D-xylopyranosyl}-16*β*-acetoxy,23*S*-hydroxyholost-7-en.

The aglycone moieties of djakonoviosides A_1_ (**2**) and B_1_ (**4**) were identical to each other, which was deduced from the coincidence of their NMR spectra (Table 3 and Appendix A). The signals corresponding to the polycyclic system of the aglycone part of **2** (Table 3, Appendix A) were close to those of **1**, indicating their identity. The signal of C-22 deduced from the common lanostane derivative HMBC correlations H-21/C-22 was observed at δ_C_ 78.3. The corresponding proton signal (H-22) was observed at δ_H_ 5.71 as a singlet. This proton was correlated in the HMBC spectrum with the signal of a quaternary carbon at δ_C_ 213.4 (C-23), corresponding to the oxo group. Therefore, a 22-hydroxy-23-oxo fragment was supposed to be present in the side chain of **2**. The coupling patterns of H-22 (s) and H-24 (dd), as well as the MS data of **2**, confirmed the supposition.

The attempt to assign a C-22 configuration in compound **2** using the modified Mosher’s method failed because the 22-*O*-MTPA esters were not formed. However, the analysis of the biogenetic background resulted in the assignment of a 22*S* configuration in djakonovioside A_1_ (**2**). Earlier, the absolute *R* configuration of the C-22 chiral center was elucidated by Mosher’s method in the glycosides of the holostane type isolated from the sea cucumber *Cladolabes schmeltzii* [29]. The same 22*R* configuration was established in the non-holostane aglycone of frondoside C by comparing its NMR data with those of model isomeric derivatives by C-22 [30]. Cucumarioside H_8_, isolated from *Eupentacta fraudatrix* and having a 16,22-epoxy fragment in the aglycone, is also characterized by the 22*R* configuration, which was established based on ROE correlation H-16/H-22 [31]. Hence, all these data suggested the same configuration, but in the case of **2** and **4** were designated as 22*S* (because of the changing of substituent seniority in comparison with earlier known glycosides). The configuration of the C-22 chiral center should be the same because of the significance of this stereocenter for the enzymatic cleavage of the side chain in the process of the biosynthesis of the nor-lanostane-type aglycones [32]. Moreover, the presence of ROE correlation H-16/H-22 in the spectrum of **4** (Appendix A) confirmed this supposition.

The molecular formula of djakonovioside A_1_ (**2**) was determined to be C_55_H_85_O_27_SNa from the [M_Na_–Na]^−^ ion peak at *m*/*z* 1209.5028 (calc. 1209.5004) and the [M_Na_–Na–H]^2−^ ion peak at *m*/*z* 604.2462 (calc. 604.2466) in the (−)HR-ESI-MS (Appendix A).

The (−)ESI-MS/MS of **2** (Appendix A) demonstrated the fragmentation of the [M_Na_–Na]^−^ ion with *m*/*z* 1209.5 giving the fragment ion peaks at *m*/*z* 1149.5 [M_Na_–Na–CH_3_COOH]^−^ and 1065.4 [M_Na_–Na–CH_3_COO–C_5_H_9_O]^−^, corresponding to the loss of the *O*-acetyl group and part of the side chain as a result of the cleavage of the C-22–C-23 bond. Fragment ion peaks at *m*/*z* 1033.4 [M_Na_–Na–MeGlc]^−^ and 889.3 [M_Na_–Na–CH_3_COO–C_5_H_9_O–MeGlc]^−^ corroborated that 3-*O*-methylglucose is a terminal residue; the ion peaks at *m*/*z* 665.2 [M_Na_–Na–Agl–H]^−^, 489.1 [M_Na_–Na–Agl–MeGlc]^−^, 357.0 [M_Na_–Na–Agl–MeGlc–Xyl]^−^, and 210.99 [M_Na_–Na–Agl–MeGlc–Xyl–Qui]^−^ confirmed the aglycone structure and the sequence of monosaccharide residues in **2**.

These data indicate that djakonovioside A_1_ (**2**) is 3*β*-*O*-[3-*O*-methyl-*β*-D-glucopyranosyl-(1→3)-*β*-D-xylopyranosyl-(1→4)-*β*-D-quinovopyranosyl-(1→2)-4-*O*-sodium sulfate-*β*-D-xylopyranosyl}-16*β*-acetoxy,22*S*-hydroxy,23-oxo-holost-7-en.

The molecular formula of djakonovioside A_2_ (**3**) was determined to be C_53_H_83_O_26_SNa from the [M_Na_–Na]^−^ ion peak at *m*/*z* 1167.4934 (calc. 1167.4899) in the (−)HR-ESI-MS (Appendix A). The aglycone moieties of djakonoviosides A_2_ (**3**) and B_4_ (**7**) were identical to each other, which was deduced from the coincidence of their NMR spectra (Table 4 and Appendix A). The signals corresponding to 18(20)-lactone at δ_C_ 180.3 (C-18) and 82.2 (C-20) indicated that the holostane-type aglycone with a 7(8)-double bond (δ_C_ 120.2 (C-7) and 146.4 (C-8)) is inherent to djakonovioside A_2_ (**3**). The signals of oxygen-substituted methine group CH-16 at δ_C_ 70.2 and δ_H_ 4.87 (dd, *J* = 7.1; 13.1 Hz) were deduced from the ^1^H,^1^H COSY spectrum, where the isolated spin system formed by the protons H_2_-15/H-16/H-17 was found. Its position was corroborated by the HMBC correlation C-15/H-16 observed in the spectrum of **3**. The characteristic ROE correlation H_3_-32/H-16 and the coupling pattern of H-16 (dd, *J*_16/17_ = 7.1 Hz) indicated an α-orientation of proton H-16. The signals of side-chain atoms were assigned beginning from C-22 (δ_C_ 70.9), deduced by the correlation H-21/C-22 in the HMBC spectrum of **3**. The corresponding H-22 signal was assigned at δ_H_ 3.96, observed as a singlet (Table 4); because of the vicinity of quaternary carbons C-20 and C-23, the latter signal was observed at δ_C_ 96.2. The values of the chemical shifts indicated the presence of a hydroxy group at C-22 along with the hemiketal fragment formed by C-23 and C-16 in the aglycone of **3**. The HMBC correlation between H-16 and C-23 observed for **3** confirmed this supposition. Only this structure of the aglycone of **3** corresponded to the chemical formula deduced from HR-ESI-MS data.

The configuration of C-22 in **3** is the same as suggested for **2**, which was deduced based on biogenetic background. The configuration of the C-23 chiral center was proposed on the basis of the observed ROE correlations and the evaluation of interatomic distances in the MM2-optimized models of the aglycones of **3** having α- or β-orientation of the hydroxyl group at C-23 (Figure 2). The distances between H-22 and H_2_-24 in the 23*α*-OH model were much less than in the 23*β*-OH model, indicating the probability of the observation of ROE correlations H-22/H-24 only in the case of 23*α*-OH. Taking into account the fact that the referred cross-peaks were observed in the ROESY spectra of **3** and **7** (Table 4 and Appendix A), the configuration of C-23 was assigned as *R*.

For the additional corroboration of the aglycone structure in djakonoviosides A_2_ (**3**) and B_4_ (**7**), the latter compound, which was isolated in sufficient amount, was acetylated to give the derivative of **7a** (Figure 3), the structure of which was established by extensive analysis of NMR and HR-ESI-MS data. The molecular formula of **7a** was determined to be C_80_H_111_O_40_SNa from the [M_Na_–Na]^−^ ion peak at *m*/*z* 1743.6353 (calc. 1743.6378) in the (−)HR-ESI-MS and the [M_Na_+Na]^+^ ion peak at *m*/*z* 1789.6051 (calc. 1789.6162) in the (+)HR-ESI-MS (Appendix A) and corresponded to the peracetylated derivative with 11 acetoxy groups. The signals in the ^13^C NMR spectrum of the aglycone moiety of **7a** (Appendix A, Appendix A) corresponded to the holostane-type polycyclic nucleus (18(20)-lactone signals at δ_C_ 178.4 (C-18) and 77.4 (C-20)) with a 7(8)-double bond (δ_C_ 120.7 (C-7) and 145.6 (C-8)). The signal of C-16 was observed at δ_C_ 77.2 being deshielded in comparison with that of the native compound 7 (δ_C-16_ 70.2). The signals of an additional double bond were observed at δ_C_ 143.2 (C-23), δ_C_ 120.6 (C-24), and δ_H_ 4.89 (d, *J* = 8.7 Hz, H-24), indicating the presence of a 16,23-ether bond and 23(24)-double bond. The latter formed as a result of intramolecular dehydratation because of the high reactivity of hemiketal hydroxyl at C-23 through the *β*-elimination mechanism under the conditions of an acetylation procedure (evaporation at t = 60 °C). The position of the double bond was confirmed by the correlations H-22/C-24, H-24/C: 22, 23, and H-25/C: 23, 24 in the HMBC spectrum of **7a**. The signals CH-22 observed at δ_C_ 71.0 and δ_H_ 5.65 (s, H-22) and the cross-peak H-22/OAc observed in the HMBC spectrum of **7a** indicated the presence of an O-acetyl group at this position. The conducting of an acetylation reaction under milder conditions (only at room temperature), followed by the registration of the MS spectra of the reaction products (Appendix A), resulted in the identification of the same compound **7a** by the characteristic ion peak at *m*/*z* 1789.6109 [M_Na_+Na]^+^ (C_80_H_111_O_40_SNa), as well as compound **7b** corresponding to molecular formula C_80_H_113_O_41_SNa by the ion peak at *m*/*z* 1807.6210 [M_Na_+Na]^+^ in the (+) HR-ESI-MS. The derivative **7b** has 11 acetoxy groups and one hydroxyl group that is obviously 23-OH because the tertiary hydroxyl is usually not exposed to acetylation. Thus, the structures of **7a** and **7b** corroborated the presence of a 23,16-hemiketal fragment in the aglycones of djakonoviosides A_2_ (**3**) and B_4_ (**7**). The triterpene nucleus with a hemiketal fragment was found for the first time among the diversity of known sea cucumber aglycones.

The (−)ESI-MS/MS of **3** (Appendix A) demonstrated the fragmentation of the [M_Na_–Na]^−^ ion with *m*/*z* 1167.5 giving the fragment ion peaks at *m*/*z* 1149.4 [M_Na_–Na−H_2_O]^−^, corresponding to the dehydrated derivative obviously formed as a result of the reaction of the *β*-elimination between C-23 and C-24; the 991.4 [M_Na_–Na–MeGlc]^−^ that confirmed 3-O-methylglucose is a terminal residue in the sugar chain, and 665.2 [M_Na_–Na–Agl–H]^−^ corroborated the aglycone structure of **3**.

These data indicate that djakonovioside A_2_ (**3**) is 3*β*-*O*-[3-*O*-methyl-*β*-D-glucopyranosyl-(1→3)-*β*-D-xylopyranosyl-(1→4)-*β*-D-quinovopyranosyl-(1→2)-4-*O*-sodium sulfate-*β*-D-xylopyranosyl}-22*S*-hydroxyholost-7-en-23*R*,16*β*-hemiketal.

Extensive analysis of the ^1^H,^1^H COSY, 1D TOCSY, HSQC, and ROESY spectra of the carbohydrate parts of djakonoviosides B_1_–B_4_ (**4**–**7**) (Table 5 and Appendix A) indicated they contain, identical to each other, monosulphated pentasaccharide chains branched at C-2 Qui2. The monosaccharide composition was determined as three xylose (Xyl1, Xyl3, and Xyl5) residues, one quinovose (Qui2) residue, and one 3-*O*-methylglucose (MeGlc4) residue. The comparison of the ^13^C NMR spectra of **4** and **1** showed the similarity of the signals of Xyl1, Xyl3, and MeGlc4, while the signal of C-2 Qui was deshielded to δ_C_ 83.1 because of the effect of glycosylation, as well as five additional signals corresponding to xylose unit that were observed in the spectrum of **4**. The positions of the glycosidic linkages established by the ROESY and HMBC correlations were the same as in the djakonoviosides of group A with additional correlation corresponding to the *β*-(1→2) glycosidic bond between C-1 Xyl5 and C-2 Qui2 (Table 5). Such a structure of a carbohydrate chain is characteristic of the cucumariosides of the A_0_ group found first in *Cucumaria japonica* [25]. The glycosides, bearing this sugar chain and isolated from *C. djakonovi*, were named the djakonoviosides of group B.

The molecular formula of djakonovioside B_1_ (**4**) was determined to be C_60_H_93_O_31_SNa from the [M_Na_–Na]^−^ ion peak at *m*/*z* 1341.5458 (calc. 1341.5427) in the (−)HR-ESI-MS (Appendix A). The (−)ESI-MS/MS of **4** (Appendix A) demonstrated the fragmentation of the [M_Na_–Na]^−^ ion with *m*/*z* 1341.5 resulting in the fragment ion peaks’ appearances at *m*/*z* 1281.5 [M_Na_–Na−CH_3_COO]^−^ and 1197.5 [M_Na_–Na−CH_3_COO–C_5_H_9_O]^−^ corresponding to the cleavage of a side chain by a C-22/C-23 covalent bond and the loss of *O*-acetyl group, 797.2 [M_Na_–Na–Agl–H]^−^ and 755.3 [M_Na_–Na–MeGlc–Xyl–Xyl]^−^. The fragmentary ion peaks formed out of the [M_Na_+Na]^+^ ion with *m*/*z* 1387.5 in the (*+*)ESI-MS/MS of **4** were observed at *m*/*z* 1267.5 [M_Na_+Na–NaHSO_4_]^+^, 1195.5 [M_Na_+Na−MeGlc+H]^+^, 861.2 [M_Na_+Na−Agl+H]^+^, 741.2 [M_Na_+Na–Agl–NaSO_4_+H]^+^, 669.2 [M_Na_+Na−Agl–MeGlc+H]^+^, and 549.3 [M_Na_+Na–Agl–MeGlc–NaSO_4_+H]^+^, corroborating the structure of djakonovioside B_1_ (**4**) established by the analysis of NMR data.

These data indicate that djakonovioside B_1_ (**4**) is 3*β*-*O*-{3-*O*-methyl-*β*-D-glucopyranosyl-(1→3)-*β*-D-xylopyranosyl-(1→4)-[(1→2)-*β*-D-xylopyranosyl]-*β*-D-quinovopyranosyl-(1→2)-4-*O*-sodium sulfate-*β*-D-xylopyranosyl}-16*β*-acetoxy,22*S*-hydroxy,23-oxo-holost-7-en.

The molecular formula of djakonovioside B_2_ (**5**) was determined to be C_58_H_91_O_29_SNa from the [M_Na_–Na]^−^ ion peak at *m*/*z* 1283.5371 (calc. 1283.5372) in the (−)HR-ESI-MS (Appendix A). The signals of the polycyclic system in the aglycone moiety of djakonovioside B_2_ (**5**) were close to those in djakonovioside A_2_ (**3**) (Table 4 and Table 6), showing the signal of C-16 at δ_C_ 69.9, and likewise, the characteristic signal of the quaternary carbon at δ_C_ 96.7 observed in the spectrum of **5** indicated the presence of a 23,16-hemiketal fragment. The correlation H-16/C-23 observed in the HMBC spectrum of **5** also corroborated the presence of this structural feature. The signals of methylene group CH_2_-22 deduced from the cross-peak H-21/C-22 in the HMBC spectrum were observed in the high field regions (δ_C_ 42.1, δ_H_ 2.39 (brd, *J* = 15.8 Hz) and 1.95 (brd, *J* = 15.1 Hz)), indicating the absence of any substituents at this position. The multiplicity of the H-22 signal corroborated its adjacency to the quaternary carbons C-20 and C-23. The *α*-orientation of H-16 was confirmed by the ROE correlation H_3_-32/H-16 and the coupling constant *J*_17/16_ = 7.0 Hz deduced from the spectra of **5** (Table 6, Appendix A). Hence, the aglycone of djakonovioside B_2_ (**5**) differed from that of djakonovioside A_2_ (**3**) by the absence of the 22-OH group and was also found for the first time in the holothurious glycosides.

The fragmentation of the [M_Na_–Na]^−^ ion at *m*/*z* 1283.5 observed in the (−)ESI-MS/MS of **5** (Appendix A) resulted in the appearance of the ion peak at *m*/*z* 1183.4 [M_Na_–Na–C_6_H_12_O]^−^ because of the loss of the fragment corresponding to the side chain after the cleavage of the covalent bonds C-20/C-22 and C-23/O. The subsequent fragmentation led to the ion peak at *m*/*z* 1007 [M_Na_–Na−C_6_H_12_O–MeGlc]^−^. The ion peak at *m*/*z* 1107.4 [M_Na_–Na–MeGlc]^−^ corresponded to the loss of the terminal 3-*O*-methylglucose unit and at *m*/*z* 797.2 [M_Na_–Na–(C_30_H_45_O_5_) Agl–H]^−^ demonstrated the loss of the aglycone moiety. The fragmentary ion peaks, formed as a result of the fragmentation of the [M_Na_+Na]^+^ ion with *m*/*z* 1329.5 in the (*+*)ESI-MS/MS of **5**, were observed at *m*/*z* 1209.6 [M_Na_+Na–NaHSO_4_]^+^, 1137.5 [M_Na_+Na–MeGlc+H]^+^, 861.2 [M_Na_+Na–Agl+H]^+^, 741.2 [M_Na_+Na–Agl–NaSO_4_]^+^, and 669.2 [M_Na_+Na–Agl–MeGlc+H]^+^.

These data indicate that djakonovioside B_2_ (**5**) is 3*β*-*O*-{3-*O*-methyl-*β*-D-glucopyranosyl-(1→3)-*β*-D-xylopyranosyl-(1→4)-[(1→2)-*β*-D-xylopyranosyl]-*β*-D-quinovopyranosyl-(1→2)-4-*O*-sodium sulfate-*β*-D-xylopyranosyl}-holost-7-en-23*R*,16*β*-hemiketal.

The molecular formula of djakonovioside B_3_ (**6**) was determined to be C_58_H_93_O_29_SNa from the [M_Na_–Na]^−^ ion peak at *m*/*z* 1285.5520 (calc. 1285.5529) in the (−)HR-ESI-MS (Appendix A). The aglycone moiety of **6**—those signals were deduced from the extensive analysis of the NMR spectra (Table 7, Appendix A)—was found to be of the lanostane type, having 18(16)-lactone (from the signals of C-16 at δ_C_ 80.0 and C-18 at δ_C_ 182.8, as well as from the distinctive signals of H-16 at δ_H_ 5.10 and H-17 at δ_H_ 2.81, both being singlets [29], and the shielded signal of C-20 (δ_C_ 72.8), compared to the same signals in the spectra of compounds **1**–**5** (at δ_C_ ∼ 82)). The 16*β*-O configuration was confirmed by the absence of coupling constant *J*_17/16_ and by the ROE correlation H-16/H-21 in the spectra of **6**. The common lanostane derivative 20*S* configuration was corroborated by the cross-peaks H-17/H-21 and H-21/H-12 observed in the ROESY spectrum of djakonovioside B_3_ (**6**). The hydroxyl group was attached to C-23 (δ_C_ 66.5) in the side chain of **6**, as in djakonovioside A (**1**) and frondoside D, which was deduced from the analyses of the COSY (H-22/H-23/H-24/H-25/H-26/H-27 correlations) and HMBC (H-21/C-22, H-22/C-23, H-24/C: 22, 23, 25, 26, 27 correlations) spectra (Table 7). The 23*S* configuration was determined on a biogenetic base. Hence, the aglycone of glycoside **6** is characterized by the combination of the 18(16)-lactone and 23-hydroxy groups, which is a new structural feature for this class of metabolites.

The (−)ESI-MS/MS of **6** (Appendix A) demonstrated that the fragmentation of the [M_Na_–Na]^−^ ion with *m*/*z* 1285.5 resulted in the fragment ion peaks at *m*/*z* 1153.5 [M_Na_–Na–Xyl]^−^, 977.4 [M_Na_–Na–Xyl–MeGlc]^−^, and 699.3 [M_Na_–Na–Xyl–MeGlc–Xyl–Qui]^−^, showing the sequential loss of monosaccharide units, confirming the carbohydrate structure. The ion peak at *m*/*z* 1141.4 [M_Na_–Na–C_8_H_17_O_2_+H]^−^ corresponded to losing part of the aglycone because of the breaking of covalent bond C-17/C-20. Generally, the MS data corroborated the structure of djakonovioside B_3_ (**5**) established by the analysis of NMR data.

These data indicate that djakonovioside B_3_ (**6**) is 3*β*-*O*-{3-*O*-methyl-*β*-D-glucopyranosyl-(1→3)-*β*-D-xylopyranosyl-(1→4)-[(1→2)-*β*-D-xylopyranosyl]-*β*-D-quinovopyranosyl-(1→2)-4-*O*-sodium sulfate-*β*-D-xylopyranosyl}-23*S*-hydroxylanost-7-en-18(16)-lactone.

The molecular formula of djakonovioside B_4_ (**7**) was determined to be C_58_H_91_O_30_SNa from the [M_Na_–Na]^−^ ion peak at *m*/*z* 1299.5307 (calc. 1299.5321) in the (−)HR-ESI-MS (Appendix A), corroborating the presence of the same aglycone as in djakonovioside A_2_ (**3**).

The ion peaks observed in (−)ESI-MS/MS of **7** (Appendix A) were derived from the fragmentation of the [M_Na_–Na]^−^ ion as a result of the cleavage of the side chain by the covalent bonds C-20/C-22 and C-23/O with *m*/*z* 1183.5 [M_Na_–Na–C_6_H_12_O_2_]^−^ and as a result of the loss of the terminal 3-O-methylglycose residue with *m*/*z* 1105.4 [M_Na_–Na–MeGlc–H]^−^. The ion peaks at *m*/*z* 797.2 [M_Na_–Na–Agl–H]^−^ and 695.3 [M_Na_–Na–Agl–NaSO_3_]^−^ corroborated the aglycone structure of **7**. In the (*+*)ESI-MS/MS of **7**, the fragmentary ion peaks were observed at *m*/*z* 1225.5 [M_Na_+Na–NaHSO_4_]^+^, 861.2 [M_Na_+Na–Agl+H]^+^, 741.2 [M_Na_+Na–Agl–NaSO_4_]^+^, 669.2 [M_Na_+Na–Agl–MeGlc+2H]^+^, corroborating the structure of djakonovioside B_4_ (**7**).

These data indicate that djakonovioside B_4_ (**7**) is 3*β*-*O*-{3-*O*-methyl-*β*-D-glucopyranosyl-(1→3)-*β*-D-xylopyranosyl-(1→4)-[(1→2)-*β*-D-xylopyranosyl]-*β*-D-quinovopyranosyl-(1→2)-4-*O*-sodium sulfate-*β*-D-xylopyranosyl}-22*S*-hydroxyholost-7-en-23*R*,16*β*-hemiketal.

### 2.2. Biosynthetic Pathways of the Aglycones of the Glycosides from C. djakonovi

It is known that holostane-type aglycones are biosynthesized via the hydroxylation of C-20 in a triterpene precursor followed by C-18 oxidation, resulting in the formation of 18(20)-lactone. When the hydroxyl groups are simultaneously present at C-16 and C-20 of the 18-carboxylated derivative, the formation of 18(16)-lactone occurred [32]. This process takes place during the biosynthesis of djakonovioside B_3_ (**6**). The hydroxylation with subsequent acetylation of C-16 preceding the oxidation of C-18 prevents the formation of 18(16)-lactone and leads to the synthesis of holostane-type aglycones in djakonoviosides A (**1**), A_1_ (**2**), B_1_ (**4**), okhotoside A_1_-1, cucumarioside A_0_-1, and frondoside D.

Taking into account the fact that the glycosides are the products of a mosaic type of biosynthesis, different biosynthetic stages can be shifted in time or change places in the sequence of the enzymatic oxidative reactions. Presumably, this occurred in the processes of formation of the aglycones with hemiketal fragments (compounds **3**, **5**, and **7**). Their precursor probably contains 16-hydroxyl, 18(20)-lactone, and obviously, a highly oxidized side chain with the 23-oxo group that is inherent for the aglycones of okhotoside A_1_-1, cucumarioside A_0_-1, djakonoviosides A_1_ (**2**), and B_1_ (**4**). The obtained data indicate that different aglycones of *C. djakonovi* glycosides are exposed to the action of the same monooxygenase during their biosynthesis. The intramolecular attack of the hydrogen of the hydroxy group at C-16 to the 23-oxo group leads to the cyclization of a 23,16-hemiketal fragment. The biogenetic network formed as a result of the biosynthesis of the aglycones found in the glycosides isolated from *C. djakonovi* is illustrated in Figure 4.

In the process of the biosynthesis of compounds **3**, **5**, and **7**, the formation of pyranose hemiketal fragments is realized as quite similar to the formation of the pyranose forms of sugars as a result of ring-chain tautomerism—a non-enzymatic intramolecular reaction occurred in an open-chain isomer, which produces more stable cyclic compounds. In our case, a similar process may lead to the formation of compounds **3**, **5**, and **7** with alpha-pyranose fragments, as shown in Figure 2. It is considered that the acetate groups are introduced through intermediate hydroxy derivatives at catalysis by O-acetyltransferases. The probable reason for the conversion into 23,16-hemiketals characteristic of *C. djakonovi* is the retardation of O-acetyltransferase action on an intermediate 16-hydroxylated derivative.

### 2.3. Bioactivity of the Glycosides and Structure–Activity Relationships

Cytotoxic activity of djakonoviosides A–B_4_ (**1**–**7**), as well as of known glycosides isolated from *C. djakonovi* (Appendix A), against erythrocytes and human breast cancer cell lines (MCF-7, T-47D, and triple negative MDA-MB-231), as well as leukemia HL-60 and embryonic kidney HEK-293, has been studied. Chitonoidoside L [33] and cisplatin were used as the positive controls (Table 8). The activity of the glycosides against MCF-7, T-47D, MDA-MB-231, and HEK293 cells were examined by MTT assay and against HL-60 by MTS assay.

Six glycosides from ten tested demonstrated potent hemolytic action, while the rest of the compounds were only moderately hemolytic because of the presence of hydroxy-groups in the side chains along with an additional 23,16-hemiketal cycle, making the aglycone more rigid (compounds **3**, **5**, and **7**), or non-holostane aglycone and hydroxy group in the side chain (glycoside **6**). The cancer cells were, as usual, less sensitive to the membranolytic action of the glycosides than of the erythrocytes. It is noticeable that compounds **1** and **2** demonstrated rather high cytotoxic activity despite both bearing the hydroxyl groups in the side chain, whose activity-decreasing action is obviously compensated for by the presence of holostane aglycones and tetrasaccharide linear monosulfated chains [4,34].

Some patterns of the structure–activity relationships were deduced from the analysis of the cytotoxic actions of the tested glycosides against cancer cell lines. Okhotoside A_1_-1 was the most active compound in the series because of the presence of a tetrasaccharide linear chain and holostane aglycone with a 7(8)-double bond and 16*β*-*O*-acetoxy group [11] without any hydroxyl groups. Cucumarioside A_0_-1 [25], having the same aglycone as okhotoside A_1_-1 (Appendix A), was slightly less active because of the presence of a branching xylose residue at C-2 Qui, while the activity of frondoside D [26], characterized by the same pentasaccharide sugar chain, was twofold decreased compared with cucumarioside A_0_-1 because of the presence of the 23-OH group in the aglycone. The possible decrease in the cytotoxicity level of djakonovioside A_1_ (**2**), featuring the aglycone identical to frondoside D, was compensated for by the presence of a tetrasaccharide chain identical to that of okhotoside A_1_-1. Generally, the activity of the djakonoviosides of group A was higher than that of the djakonoviosides of group B containing the same aglycones, indicating the significance of a linear sugar chain containing four monosaccharide residues for the demonstration of membranolytic action by the glycosides. All the compounds containing hemiketal fragments, djakonovioside A_2_ (**3**), B_2_ (**5**), and B_4_ (**7**), were inactive against cancer cells, as was djakonovioside B_3_ (**6**), having non-holostane aglycone and a hydroxyl group in the side chain. All these patterns are in good agreement with the structure–activity relationships established earlier for the triterpene glycosides of sea cucumbers [34].

As regards the sensitivity of cancer cell lines to glycoside action, HL-60 and MDA-MB-231 (triple-negative breast cancer) were exposed to cytotoxic action to the greatest extent, while the MCF-7 cell line was more resilient.

For further investigation of glycoside action on colony formation and migration, the triple-negative breast cancer MDA-MB-231 cells were used, and the most active okhotoside A_1_-1 and cucumarioside A_0_-1, as well as djakonoviosides A (**1**) and A_1_ (**2**), were selected. The glycosides did not lose cytotoxicity over time, and their effects increased after 48 and 72 h of incubation with the MDA-MB-231 cells (Figure 5). For example, cucumarioside A_0_-1 demonstrated an almost twofold increase in the activity (EC_50_ 6.04 2.45 and 2.19 μM) after 48 and 72 h of exposition (Figure 5C).

Clonogenic analysis is commonly used to study the influence of cytotoxic compounds on the survival and division of the cells and their ability to form colonies. The investigation of the action of selected glycosides in non-cytotoxic concentrations on the formation and growth of colonies of MDA-MB-231 cells demonstrated that the maximum inhibitory effect (70.76 ± 0.13% of the control) was observed for okhotoside A_1_-1 at a concentration of 0.5 μM (Figure 6). Cucumarioside A_0_-1 significantly inhibited the growth of colonies at all concentrations studied; the maximal blockage of the formation and growth of colonies by 43.54 ± 6.07% of the control was observed at a concentration of 0.5 μM. Djakonovosides A (**1**) and A_1_ (**2**) showed a dose-dependent effect: a statistically significant inhibition of colony growth by 41.39 ± 3.16% and 19.24 ± 0.25% of the control was observed for **1** and **2**, respectively, at the maximum concentration of 2 μM.

Migration of tumor cells plays a crucial role in the process of metastases growth, so the search for substances capable of inhibiting this process is very important. The ability of the glycosides to inhibit migration of MDA-MB-231 cells was tested in vitro by wound scratch migration assay. In the control group, MDA-MB-231 cells completely close migration to the wound area at 24 h (Figure 7A). The statistically significant effects of the glycosides on migration were also observed after 24 h of incubation.

Compound **1** showed a maximum blockage of the migration by 79.92 ± 0.27% in relation to control at a concentration of 1 μM (Figure 7A,B). A dose-dependent inhibitory effect was observed for **2** and okhotoside A_1_-1 (Figure 7C,E). Djakonovioside A_1_ (**2**) inhibited the migration of tumor cells by 79.52 ± 9.12% as compared to control at a concentration of 2 μM after 24 h incubation, while okhotoside A_1_-1 inhibited the migration by 74.76 ± 4.41%, already at the minimum studied concentration of 0.05 µM. Cucumarioside A_0_-1 strongly affected cell migration at all concentrations with approximately equal intensity (Figure 7D). Thus, at the maximum concentration of 0.5 µM, the blocking of migration was 84.62 ± 2.89% in comparison with control, and at the minimum concentration of 0.05 µM, this value was 83.61 ± 0.72%.

Generally, some of the investigated glycosides demonstrated encouraging action against breast cancer cells, suppressing their viability and inhibiting the formation and growth of colonies and the ability of the cells to migrate of the most aggressive triple-negative MDA-MB-231 line of breast cancer.

## 3. Materials and Methods

### 3.1. General Experimental Procedures

Specific rotation was measured on a PerkinElmer 343 Polarimeter (PerkinElmer, Waltham, MA, USA); NMR spectra were registered on a Bruker AMX 500 (Bruker BioSpin GmbH, Rheinstetten, Germany) (500.12/125.67 MHz (^1^H/^13^C) spectrometer; ESI MS (positive and negative ion modes) spectra were registered on an Agilent 6510 Q-TOF apparatus (Agilent Technology, Santa Clara, CA, USA) with a sample concentration of 0.01 mg/mL; HPLC was conducted on an Agilent 1260 Infinity II equipped with a differential refractometer (Agilent Technology, Santa Clara, CA, USA); columns used Discovery Ascentis RP-Amide (10 × 250 mm, 5 µm) (Supelco, Bellefonte, PA, USA) and Phenomenex Synergi Fusion RP (10 × 250 mm, 5 µm) (Phenomenex, Torrance, CA, USA).

### 3.2. Animals and Cells

The specimens of sea cucumber *Cucumaria djakonovi* (family Cucumariidae; order Dendrochirotida) were collected in the Avacha Gulf near Starichkov’s Island in July 2007 by scuba diving from a depth of 14–15 m. The taxonomic identification of the animals was performed by Dr. V.G. Stepanov. A voucher specimen is kept in the Pacific Institute of Geography, Kamchatka Branch, Petropavlovsk-Kamchatsky, Russia.

Human erythrocytes were purchased from the Station of Blood Transfusion in Vladivostok. Human promyeloblast cell line HL-60 CCL-240, human embryonic kidney HEK-293 CRL-1573^TM^ cell line, human breast cancer cell lines T-47D HTB-133^TM^, MCF-7 HTB-22^TM^, and MDA-MB-231 CRM-HTB-26^TM^ were received from ATCC (Manassas, VA, USA). HL-60 and T-47D cell lines were cultured in a medium of RPMI with 1% penicillin/streptomycin (Biolot, St. Petersburg, Russia) and 10% fetal bovine serum (FBS) (Biolot, St. Petersburg, Russia). The HEK293 cell line was cultured in a medium of DMEM (Gibco Dulbecco’s modified Eagle medium) with 1% penicillin/streptomycin sulfate (Biolot, St. Petersburg, Russia) and 10% fetal bovine serum (FBS) (Biolot, St. Petersburg, Russia). The cells of the MCF-7 and MDA-MB-231 lines were cultured in MEM (minimum essential medium) with 1% penicillin/streptomycin sulfate (Biolot, St. Petersburg, Russia) and with fetal bovine serum (Biolot, St. Petersburg, Russia) to a final concentration of 10%.

### 3.3. Extraction and Isolation

The body walls and tentacles of the sea cucumbers were minced and extracted twice with refluxing 70% EtOH. The dry weight of raw material after extraction was 663.5 g. The combined extracts were concentrated to dryness in vacuum, dissolved in H2O, and chromatographed on a Polychrom-1 column (powdered Teflon, Biolar, Latvia). The first elution of the inorganic salts and impurities with H_2_O, followed by the elution of glycosides with 55% acetone, produced 1379 mg of a crude glycoside fraction. Its initial separation was achieved by using the chromatography on Si gel columns (CC) with the stepped gradient of the system of eluents CHCl_3_/EtOH/H_2_O in ratios 100:50:4, 100:75:10, 100:100:17, and 100:125:25 resulting in the obtaining of five fractions. Fractions I and II were repeatedly subjected to CC with the system of eluents CHCl_3_/EtOH/H_2_O (100:75:10), each for subsequent purification that resulted in obtaining subfractions 1 (244 mg) and 2 (640 mg), respectively. The HPLC of subfraction 1 on a reversed-phase column, Supelco Ascentis RP-Amide (10 × 250 mm), with MeOH/H_2_O/NH_4_OAc (1 M water solution) and ratio (57/42/1), as mobile phase, produced four fractions. The rechromatography of two of them on a Phenomenex Synergi Fusion (10 × 250 mm) column with MeOH/H_2_O/NH_4_OAc (1 M water solution) (68/30/2) as the mobile phase resulted in the isolation of djakonoviosides A (**1**) (5.1 mg, Rt 12.98 min) and A_1_ (**2**) (5.2 mg, Rt 14.52 min), as well as of the known okhotoside A_1_-1 (12.8 mg, Rt 16.36 min). Subfraction 2 was subjected to HPLC on a Phenomenex Synergi Fusion (10 × 250 mm) column with MeOH/H_2_O/NH_4_OAc (1 M water solution) (65/33/2) as the mobile phase to produce fractions 2.1–2.5 and individual cucumarioside A_0_-1 (111.6 mg, Rt 18.45 min). All the subfractions, 2.1–2.5, were repeatedly subjected HPLC on the same column but used different ratios of the CH_3_CN/H_2_O/NH_4_OAc (1 M water solution) solvent system as the mobile phase. The rechromatography of subfraction 2.4 with the ratio of solvents (40/58/2) led to the isolation of 54.0 mg of djakonovioside B_1_ (**4**) (Rt 11.52 min). The separation of subfraction 2.3 with the ratio (36/62/2) produced frondoside D (8.0 mg, Rt 16.89 min), djakonovioside B_2_ (**5**) (7.0 mg, Rt 15.52 min), and djakonovioside A_2_ (**3**) (3.2 mg, Rt 14.03 min). In the same way, the HPLC of subfraction 2.2 resulted in the isolation of djakonovioside B_3_ (**6**) (4.6 mg, Rt 12.75 min). Subfraction 2.1 was separated with the mobile phase having the ratio of solvents (35/63/2) that allowed us to obtain 65.4 mg of djakonovioside B_4_ (**7**) (Rt 13.02 min).

#### 3.3.1. Djakonovioside A (**1**)

Colorless powder; [α]_D_^20^−19° (*c* 0.1, H_2_O). NMR: Table 1 and Table 2, Appendix A. (−)HR-ESI-MS *m*/*z*: 1195.5209 (calc. 1195.5212) [M_Na_–Na]^−^; (−)ESI-MS/MS *m*/*z*: 1049.4 [M_Na_–Na−C_5_H_11_O–CH_3_COO]^−^, 1033 [M_Na_–Na−CH_3_COO–SO_3_Na]^−^, 989 [M_Na_–Na−−CH_3_COO–SO_3_Na–CO_2_]^−^, 975.4 [M_Na_–Na−CO_2_−C_7_H_13_O_5_ (MeGlc)]^−^, 741.3 [M_Na_–Na−C_7_H_13_O_5_ (MeGlc)−C_5_H_8_O_4_ (Xyl)−C_6_H_10_O_4_ (Qui)]^−^, 681.2 [M_Na_–Na−(C_32_H_49_O_5_) Agl−H]^−^.

#### 3.3.2. Djakonovioside A_1_ (**2**)

Colorless powder; [α]_D_^20^−23° (*c* 0.1, H_2_O). NMR: Table 3 and Appendix A. (−)HR-ESI-MS *m*/*z*: 1209.5028 (calc. 1209.5004) [M_Na_–Na]^−^, 604.2462 (calc. 604.2466) [M_Na_–Na–H]^2−^; (−)ESI-MS/MS *m*/*z*: 1149.5 [M_Na_–Na−CH_3_COOH]^−^, 1065.4 [M_Na_–Na−CH_3_COO–C_5_H_9_O]^−^, 1033.4 [M_Na_–Na−C_7_H_13_O_5_ (MeGlc)]^−^, 889.3 [M_Na_–Na−CH_3_COO–C_5_H_9_O–C_7_H_13_O_5_ (MeGlc)]^−^, 665.2 [M_Na_–Na−C_32_H_47_O_7_ (Agl)–H]^−^, 489.1 [M_Na_–Na−C_32_H_47_O_7_ (Agl)–C_7_H_13_O_5_ (MeGlc)]^−^, 357.0 [M_Na_–Na−C_32_H_47_O_7_ (Agl)–C_7_H_13_O_5_ (MeGlc)–C_5_H_8_O_4_ (Xyl)]^−^, 210.99 [M_Na_–Na−C_32_H_47_O_7_ (Agl)–C_7_H_13_O_5_ (MeGlc)–C_5_H_8_O_4_ (Xyl)–C_6_H_10_O_4_ (Qui)]^−^.

#### 3.3.3. Djakonovioside A_2_ (**3**)

Colorless powder; [α]_D_^20^−28° (*c* 0.1, H_2_O). NMR: Table 4 and Appendix A. (−)HR-ESI-MS *m*/*z*: 1167.4934 (calc. 1167.4899) [M_Na_–Na]^−^; (−)ESI-MS/MS *m*/*z*: 1149.4 [M_Na_–Na−H_2_O]^−^, 991.4 [M_Na_–Na−C_7_H_12_O_5_ (MeGlc)]^−^, 665.2 [M_Na_–Na−C_30_H_45_O_6_ (Agl)–H]^−^.

#### 3.3.4. Djakonovioside B_1_ (**4**)

Colorless powder; [α]_D_^20^−34° (*c* 0.1, H_2_O). NMR: Table 4 and Appendix A. (−)HR-ESI-MS *m*/*z*: 1341.5458 (calc. 1341.5427) [M_Na_–Na]^−^; (−)ESI-MS/MS *m*/*z*: 1281.5 [M_Na_–Na−CH_3_COO]^−^, 1197.5 [M_Na_–Na−CH_3_COO–C_5_H_9_O]^−^, 797.2 [M_Na_–Na−(C_32_H_47_O_7_) Agl–H]^−^, 755.3 [M_Na_–Na−C_7_H_13_O_5_ (MeGlc)–C_5_H_4_O_8_ (Xyl)–C_5_H_4_O_8_ (Xyl)–H]^−^; (*+*)ESI-MS/MS *m*/*z*: 1267.5 [M_Na_+Na−NaHSO_4_]^+^, 1195.5 [M_Na_+Na−C_7_H_13_O_6_ (MeGlc)+H]^+^, 861.2 [M_Na_+Na−C_32_H_47_O_6_ (Agl)+H]^+^, 741.2 [M_Na_+Na−C_32_H_47_O_6_ (Agl)–NaSO_4_+H]^+^, 669.2 [M_Na_+Na−C_32_H_47_O_6_ (Agl)–C_7_H_13_O_6_ (MeGlc)+H]^+^, 549.3 [M_Na_+Na−C_32_H_47_O_6_ (Agl)–C_7_H_13_O_6_ (MeGlc)–NaSO_4_+H]^+^.

#### 3.3.5. Djakonovioside B_2_ (**5**)

Colorless powder; [α]_D_^20^−29° (*c* 0.1, H_2_O). NMR: Table 5 and Appendix A. (−)HR-ESI-MS *m*/*z*: 1283.5371 (calc. 1283.5372) [M_Na_–Na]^−^; (−)ESI-MS/MS *m*/*z*: 1183.4 [M_Na_–Na−C_6_H_12_O]^−^, 1107.4 [M_Na_–Na–C_7_H_12_O_5_ (MeGlc)]^−^, 1007 [M_Na_–Na−C_6_H_12_O–C_7_H_12_O_5_ (MeGlc)]^−^, 797.2 [M_Na_–Na–C_30_H_45_O_5_ (Agl)–H]^−^; (*+*)ESI-MS/MS *m*/*z*: 1329.5 [M_Na_+Na]^+^, 1209.6 [M_Na_+Na−NaHSO_4_]^+^, 1137.5 [M_Na_+Na−C_7_H_13_O_6_ (MeGlc)+H]^+^, 861.2 [M_Na_+Na–C_30_H_45_O_4_ (Agl)+H]^+^, 741.2 [M_Na_+Na–C_30_H_45_O_4_ (Agl)–NaSO_4_]^+^, 669.2 [M_Na_+Na–C_30_H_45_O_4_ (Agl)–C_7_H_12_O_6_ (MeGlc)+H]^+^.

#### 3.3.6. Djakonovioside B_3_ (**6**)

Colorless powder; [α]_D_^20^−31° (*c* 0.1, H_2_O). NMR: Table 7 and Appendix A. (−)HR-ESI-MS *m*/*z*: 1285.5520 (calc. 1285.5529) [M_Na_–Na]^−^; (−)ESI-MS/MS *m*/*z*: 1153.5 [M_Na_–Na−C_5_H_4_O_8_ (Xyl)]^−^, 1141.4 [M_Na_–Na−C_8_H_17_O_2_+H]^−^, 977.4 [M_Na_–Na−C_5_H_4_O_8_ (Xyl)–C_7_H_13_O_5_ (MeGlc)+H]^−^, 699.3 [M_Na_–Na−C_5_H_4_O_8_ (Xyl)–C_7_H_13_O_5_ (MeGlc)–C_5_H_4_O_8_ (Xyl)–C_6_H_10_O_4_ (Qui)]^−^.

#### 3.3.7. Djakonovioside B_4_ (**7**)

Colorless powder; [α]_D_^20^−36° (*c* 0.1, H_2_O). NMR: Appendix A. (−)HR-ESI-MS *m*/*z*: 1299.5307 (calc. 1299.5321) [M_Na_–Na]^−^; (−)ESI-MS/MS *m*/*z*: 1183.5 [M_Na_–Na−C_6_H_12_O_2_–H]^−^, 1105.4 [M_Na_–Na−C_7_H_13_O_6_ (MeGlc)–H]^−^, 797.2 [M_Na_–Na−C_30_H_45_O_6_ (Agl)–H]^−^, 695.3 [M_Na_–Na−C_30_H_45_O_6_ (Agl)–NaSO_3_]^−^; (*+*)ESI-MS/MS *m*/*z*: 1225.5 [M_Na_+Na−NaHSO_4_]^+^, 861.2 [M_Na_+Na−C_30_H_45_O_5_ (Agl)+H]^+^, 741.2 [M_Na_+Na−C_30_H_45_O_5_ (Agl)–NaSO_4_]^+^, 669.2 [M_Na_+Na−C_30_H_45_O_5_ (Agl)–C_7_H_13_O_6_ (MeGlc)+2H]^+^.

### 3.4. Acetylation of Djakonovioside B_4_ (**7**)

Glycoside **7** (4 mg) was dissolved in 1 mL of a mixture of absolute pyridine and acetyl anhydride (1:1) and kept at room temperature for 12 h. The mixture was evaporated in vacuo (at 60 °C) to obtain acetylated derivative **7a** and dissolved in C_5_D_5_N for NMR spectra registration. Derivative **7b** was obtained by the same procedure, excluding the heating.

### 3.5. Cytotoxic Activity (MTT Assay) (for HEK293, MCF-7, T-47D, and MDA-MB-231 Cells)

All the substances were tested in concentrations from 0.1 µM to 50 µM. Cisplatin was used as positive control. The cell suspension (180 µL) and solutions (20 µL) of tested glycosides in different concentrations were injected in wells of 96-well plates (MCF-7, T-47D, MDA-MB-231, and HEK293—7 × 103 cells/well) and incubated at 37 °C for 24 h in an atmosphere with 5% CO_2_. Glycosides **1** and **2**, cucumarioside A_0_-1, and okhotoside A_1_-1 at concentrations of 1.25–10.0 μM were incubated with MDA-MB-231 cells at 37 °C for 24, 48, and 72 h in an atmosphere with 5% CO_2_. After incubation, the glycosides with medium were replaced by 100 µL of fresh medium. Then, 10 µL of MTT (3-(4,5-dimethylthiazol-2-yl)-2,5-diphenyltetrazolium bromide) (Sigma-Aldrich, St. Louis, MO, USA) stock solution (5 mg/mL) was added to each well, followed by incubation of the microplate for 4 h. After this procedure, 100 µL of SDS-HCl solution (1 g SDS/10 mL d-H2O/17 µL 6 N HCl) was added to each well and incubated for 18 h. The absorbance of the converted dye formazan was determined with a Multiskan FC microplate photometer (Thermo Fisher Scientific, Waltham, MA, USA) at 570 nm. Cytotoxic activity of the tested glycosides was calculated as a concentration that caused 50% cell metabolic activity inhibition (IC50). The experiments were conducted in triplicate; *p* < 0.05.

### 3.6. Cytotoxic Activity (MTS Assay) (for HL-60)

The cells of the HL-60 line (10 × 10^3^/200 µL) were placed in 96-well plates at 37 °C for 24 h in a 5% CO_2_ incubator and, then, treated with tested glycosides and cisplatin as positive control at concentrations between 0.1 and 50 µM for an additional 24 h of incubation. Then, the cells were incubated with 10 µL MTS ([3-(4,5-dimethylthiazol-2-yl)-5-(3-carboxymethoxyphenyl)-2-(4-sulfophenyl)-2H-tetrazolium) for 4 h, and the absorbance in each well was determined at 490/630 nm with a PHERA star FS plate reader (BMG Labtech, Ortenberg, Germany). The experiments were conducted in triplicate. The results were presented as the percentage of inhibition that produced a reduction in absorbance after tested glycosides treatment compared to the nontreated cells (negative control); *p* < 0.01.

### 3.7. Hemolytic Activity

Erythrocytes were obtained from human blood (AB(IV) Rh+) by centrifugation with phosphate-buffered saline (PBS) (pH 7.4) at 4 °C for 5 min by 450 g on a LABOFUGE 400R centrifuge (Heraeus, Hanau, Germany) three times. Then, the erythrocytes residue was resuspended in ice cold phosphate saline buffer (pH 7.4) to a final optical density of 1.5 at 700 nm and kept on ice. For the hemolytic assay, 180 µL of erythrocyte suspension was mixed with 20 µL of test compound solution, as well as control, chitonoidoside L [33], in V-bottom 96-well plates. After 1 h of incubation at 37 °C, the plates were exposed to centrifugation for 10 min at 900 g in a laboratory LMC-3000 centrifuge (Biosan, Riga, Latvia). Then, 100 µL of supernatant was carefully decanted and transferred into new flat plates. The values of the lysis of the erythrocytes were found by measuring the concentration of hemoglobin in the supernatant with a Multiskan FC microplate photometer (Thermo Fisher Scientific, Waltham, MA, USA); λ = 570 nm. The effective dose causing 50% hemolysis of erythrocytes (ED50) was calculated with a SigmaPlot 10.0 computer program. All the experiments were carried out in triple repetitions; *p* < 0.01.

### 3.8. Colony Formation Assay

The influence of glycosides on the proliferation of MDA-MB-231 cells was analyzed by clonogenic assay [35]. Briefly, MDA-MB-231 cells were cultured on 6-well plates at a density of 1 × 10^3^ cells per well in control media (MEM media, 10% FBS, 10,000 U/mL of penicillin and 10,000 μg/mL of streptomycin) or in media supplemented with different concentrations of glycosides. Cells were incubated for one week at 37 °C with 5% CO_2_ until the cells in the control plates formed colonies that were visible to the eye and were of a substantial size (at least 50 cells per colony). For fixation and staining, the media were removed and the cells were washed twice with PBS. The colonies were fixed with methanol for 25 min, then washed with PBS and stained with 0.5% crystal violet solution for 25 min at room temperature. The plates were then washed with water and air dried.

### 3.9. Wound Scratch Migration Assay

To analyze the influence of the tested compounds on tumor cell migration, MDA-MB-231 cells attached to the plate’s plastic bottom were separated by a silicone insert from special migration plates (Culture-insert 2 Well 24, ibiTreat); then, the insert was removed, leaving a gap of 500 ± 50 μm (according to the manufacturer’s data) between the cells. The cells were washed twice by PBS to remove cell debris and floating cells and loaded with a fluorescent probe, a (5,6)-carboxyfluorescein succinimidyl ester (CFDA SE) dye (LumiTrace CFDA SE kit, Lumiprobe, Moscow, Russia). The initial solution of CFDA SE at a concentration of 5 mM in DMSO was dissolved in PBS to prepare a 10 μM solution and was added to the cells for 5 min at 37 °C; then, the cells were washed twice with PBS, and fresh culture medium was added. After that, the cells were treated with various concentrations of glycosides and left for 8 and 24 h. Cells treated with culture medium only were used as control. Cell migration into the wound area was then observed under a fluorescence microscope (MIB-2-FL, LOMO, Saint Peterburg, Russia) with an objective ×10 magnification.

## 4. Conclusions

At the first stage of investigation of the glycosidic composition of the sea cucumber *Cucumaria djakonovi,* two fewer polar fractions containing monosulfated tetra- and pentaosides have been studied that resulted in the isolation of seven new djakonoviosides, A–B_4_ (**1**–**7**), and three known glycosides found earlier in other representatives of the *Cucumaria* genus. The analysis of the structural peculiarities of isolated compounds revealed five different aglycones, four of them found for the first time, and two types of carbohydrate chains common for the glycosides of the *Cucumaria* species. Therefore, the trend that was disclosed earlier concerning the species specificity of the set of aglycones, as well as the genus-specific raw carbohydrate chains of the glycosides of the sea cucumbers belonging to genus *Cucumaria* [6], was confirmed by the structures of compounds from *C. djakonovi*. Unprecedented structural features of the glycosides **3**, **5**, and **7** consist of the presence of a pyranose 23,16-hemiketal cycle, formed similarly to the appearance of pyranose forms of sugars. Probably, it is connected with the retardation of acylation of a free hydroxyl at C-16 by the corresponding O-acetyltransferases in *Cucumaria djakonovi.*

Noticeably, the known okhotoside A_1_-1 and cucumarioside A_0_-1 isolated earlier from *C. djakonovi* demonstrated promising effects against the most aggressive triple-negative MDA-MB-231 cell line of breast cancer, significantly inhibiting the formation and growth of colonies and the migration of cells.

## Figures and Tables

**Figure 1 ijms-24-11128-f001:**
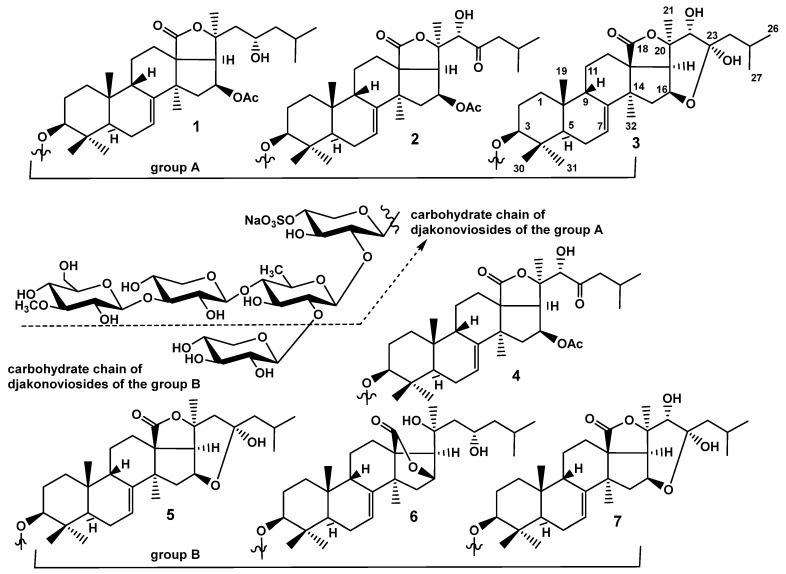
Chemical structures of glycosides isolated from *Cucumaria djakonovi*: **1**—djakonovioside A; **2**—djakonovioside A_1_; **3**—djakonovioside A_2_; **4**—djakonovioside B_1_; **5**—djakonovioside B_2_; **6**—djakonovioside B_3_; and **7**—djakonovioside B_4_.

**Figure 2 ijms-24-11128-f002:**
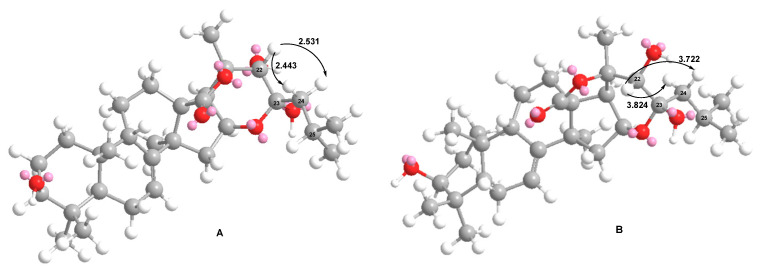
MM2-optimized models of the aglycone of djakonoviosides A_2_ (**3**) and B_4_ (**7**) with 23*α*-OH (**A**) and 23*β*-OH (**B**) and interatomic distances between H-22 and H_2_-24 in Å.

**Figure 3 ijms-24-11128-f003:**
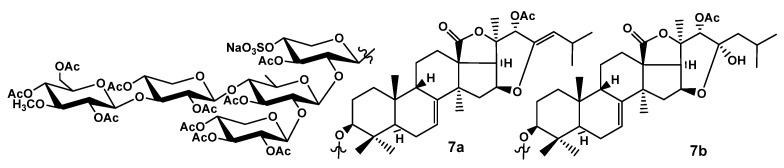
Chemical structure of acetylated derivatives **7a** and **7b** of djakonovioside B_4_ (**7**).

**Figure 4 ijms-24-11128-f004:**
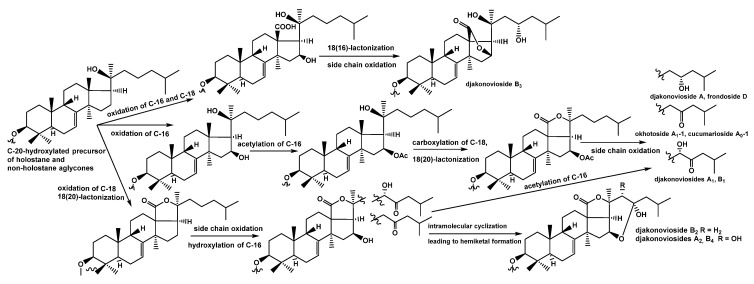
The scheme of biosynthesis of holostane and non-holostane aglycones of the glycosides of *C. djakonovi*.

**Figure 5 ijms-24-11128-f005:**
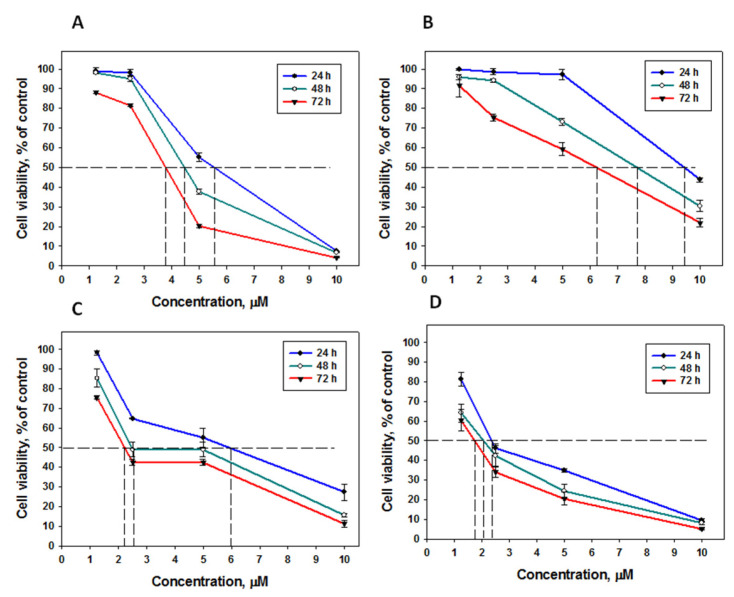
Cytotoxic effect of the glycosides: (**A**)—djakonovioside A (**1**) (EC_50_ 5.89, 4.45, and 3.77 μM for 24, 48, and 72 h, respectively); (**B**)—djakonovioside A_1_ (**2**) (9.64, 7.33, and 6.25 μM for 24, 48, and 72 h, respectively), (**C**)—cucumarioside A_0_-1 (EC_50_ 6.04, 2.45, and 2.19 μM for 24, 48, and 72 h, respectively), and (**D**)—okhotoside A_1_-1 (2.34, 2.05, and 1.73 μM for 24, 48, and 72 h, respectively) on breast cancer cells MDA-MB-231 for 24 h, 48 h, and 72 h. All experiments were carried out in triplicate. The data are presented as mean ± SEM.

**Figure 6 ijms-24-11128-f006:**
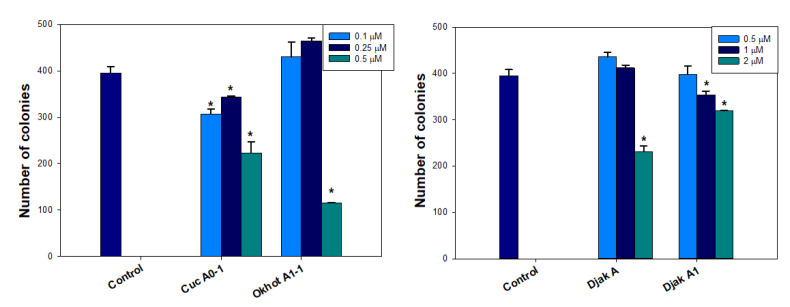
The number of MDA-MB-231 cell colonies under treatment with different concentrations of cucumarioside A_0_-1, okhotoside A_1_-1, and djakonoviosides A (**1**) and A_1_ (**2**). Image J 1.52 software was used to count the cell colonies. Data are presented as means ± SEM. * *p* value ≤ 0.05 considered significant.

**Figure 7 ijms-24-11128-f007:**
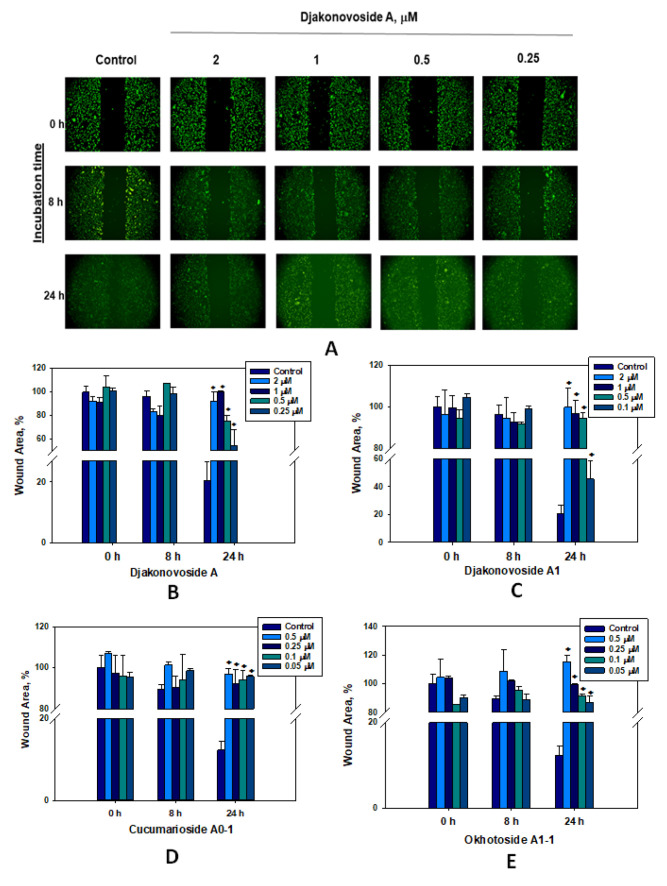
Migration of MDA-MB-231 cells into wound areas observed with an MIB-2-FL fluorescence microscope of ten-fold magnification: (**A**)—0, 8, and 24 h after treatment with different concentrations of djakonovioside A (**1**). Cell migration into wound areas processed by Image J 1.52 software: (**B**)—0, 8, and 24 h after treatment with 0.25, 0.5, 1.0, and 2.0 μM of djakonovioside A (**1**); (**C**)—0, 8, and 24 h after treatment with 0.1, 0.5, 1.0, and 2.0 μM of djakonovioside A_1_ (**2**); (**D**)—0, 8, and 24 h after treatment with 0.05, 0.1, 0.25, and 0.5 μM of cucumarioside A_0_-1; (**E**)—0, 8, and 24 h after treatment with 0.05, 0.1, 0.25, and 0.5 μM of okhotoside A_1_-1. Data are presented as means ± SEM. * *p* value ≤ 0.05 considered significant.

**Table 1 ijms-24-11128-t001:** ^13^C and ^1^H NMR chemical shifts, HMBC and ROESY correlations of carbohydrate moiety of djakonovioside A (**1**).

Atom	δ_C_ mult. *^a, b, c^*	δ_H_ mult. (*J* in Hz) *^d^*	HMBC	ROESY
Xyl1 (1→C-3)				
1	105.0 CH	4.75 d (6.9)	C: 3	H-3; H-3, 5 Xyl1
2	**83.3** CH	4.05 t (8.2)	C: 1 Qui2; C: 1 Xyl1	H-1 Qui2
3	75.6 CH	4.34 t (8.2)	C: 2, 4 Xyl1	
4	*75.9* CH	5.15 m		
5	64.1 CH_2_	4.83 brdd (5.7; 11.4)	C: 1 Xyl1	
		3.79 t (11.4)	C: 1, 4 Xyl1	
Qui2 (1→2Xyl1)				
1	105.2 CH	5.13 d (7.6)	C: 2 Xyl1	H-2 Xyl1; H-3, 5 Qui2
2	76.3 CH	4.02 t (8.8)	C: 1, 3 Qui2	
3	75.4 CH	4.11 t (8.8)	C: 2, 4 Qui2	
4	**85.7** CH	3.65 t (8.8)	C: 1 Xyl3; C: 3, 5 Qui2	H-1 Xyl3
5	71.6 CH	3.77 m	C: 4 Qui2	H-1 Qui2
6	17.8 CH_3_	1.73 d (5.8)	C: 4, 5 Qui2	H-4 Qui2
Xyl3 (1→4Qui2)				
1	105.0 CH	4.86 d (7.5)	C: 4 Qui2	H-4 Qui2; H-3, 5 Xyl3
2	73.2 CH	3.99 t (8.2)	C: 1, 3 Xyl3	
3	**87.3** CH	4.13 t (8.2)	C: 1 MeGlc4; C: 2, 4 Xyl3	H-1 MeGlc4; H-1, 5 Xyl3
4	68.9 CH	4.04 m	C: 5 Xyl3	
5	66.3 CH_2_	4.21 dd (5.4; 11.5)	C: 1, 3, 4 Xyl3	
		3.63 t (11.5)	C: 3, 4 Xyl3	
MeGlc4 (1→3Xyl3)				
1	105.3 CH	5.28 d (8.1)	C: 3 Xyl3	H-3 Xyl3; H-3, 5 MeGlc4
2	74.9 CH	4.00 t (8.8)	C: 1 MeGlc4	
3	87.8 CH	3.72 t (8.8)		H-1, 5 MeGlc4; OMe
4	70.5 CH	4.14 t (8.8)	C: 3, 5, 6 MeGlc4	
5	78.1 CH	3.95 m	C: 6 MeGlc4	H-1, 3 MeGlc4
6	62.0 CH_2_	4.45 dd (3.4; 11.5)	C: 4, 5 MeGlc4	
		4.26 dd (5.4; 11.5)	C: 4, 5 MeGlc4	
OMe	60.5 CH_3_	3.87 s	C: 3 MeGlc4	

*^a^* Recorded at 125.67 MHz in C_5_D_5_N. *^b^* Bold = interglycosidic positions. *^c^* Italic = sulfate position. *^d^* Recorded at 500.12 MHz in C_5_D_5_N. Multiplicity by 1D TOCSY. The original spectra of **1** are provided as Appendix A.

**Table 2 ijms-24-11128-t002:** ^13^C and ^1^H NMR chemical shifts, HMBC and ROESY correlations of aglycone moiety of djakonovioside A (**1**).

Position	δ_C_ mult. *^a^*	δ_H_ mult. (*J* in Hz) *^b^*	HMBC	ROESY
1	36.0 CH_2_	1.44 m		
		1.40 m		
2	26.9 CH_2_	2.10 m		
		1.89 m		
3	89.0 CH	3.27 dd (4.0; 11.6)	C: 4, 30, 31, C:1 Xyl1	H-1, H-5, H-31, H1-Xyl1
4	39.4 C			
5	47.7 CH	1.03 m	C: 4, 6, 10, 30, 31	H-3, H-31
6	23.2 CH_2_	2.06 m	C: 7, 8	
7	120.2 CH	5.65 m	C: 5	H-15, H-32
8	145.7 C			
9	47.1 CH	2.48 brd (14.1)		
10	35.4 C			
11	22.6 CH_2_	1.80 m	C: 10	
		1.50 m	C: 10	
12	31.5 CH_2_	2.22 brdd (5.5; 8.6)	C: 9, 13, 18	H-21
		2.02 d (8.6; 14.1)	C: 9, 11, 13, 17, 18	
13	58.9 C			
14	48.5 C			
15	44.1 CH_2_	2.70 dd (6.7; 12.3)	C: 13, 14, 17, 32	H-7, H-32
		1.75 d (6.1)	C: 14, 16, 32	
16	75.2 CH	5.96 dd (7.8; 16.8)	C: 13, 20, OAc	H-32
17	55.4 CH	2.93 d (8.9)	C: 12, 13, 18, 21	H-21, H-32
18	179.7 C			
19	23.8 CH_3_	1.24 s	C: 1, 5, 9	H-1, H-6, H-9, H-30
20	85.3 C			
21	30.0 CH_3_	1.96 s	C: 17, 20, 22	H-12, H-17, H-22, H-23
22	47.0 CH_2_	2.67 d (14.5)	C: 20, 21	
		2.28 dd (10.2; 14.3)	C: 17, 20, 21, 23, 24	H-21
23	65.9 CH	4.08 m		
24	48.9 CH_2_	1.67 m		H-26
		1.23 m		
25	24.3 CH	2.16 m		
26	23.9 CH_3_	0.96 d (6.6)	C: 24, 25, 27	H-24
27	21.5 CH_3_	1.01 d (6.6)	C: 24, 25, 26	H-23
30	17.1 CH_3_	1.15 s	C: 3, 4, 5, 31	H-2, H-6, H-19, H-31
31	28.5 CH_3_	1.30 s	C: 3, 4, 5, 30	H-3, H-5, H-6, H-30, H-1 Xyl1
32	32.4 CH_3_	1.10 s	C: 8, 13, 14, 15	H-7, H-12, H-15, H-16, H-17
OCOCH_3_	169.5 C			
OCOCH_3_	21.3 CH_3_	2.04 s	C: 16, OAc	

*^a^* Recorded at 125.67 MHz in C_5_D_5_N. *^b^* Recorded at 500.12 MHz in C_5_D_5_N. The original spectra of **1** are provided as Appendix A.

**Table 3 ijms-24-11128-t003:** ^13^C and ^1^H NMR chemical shifts, HMBC and ROESY correlations of aglycone moiety of djakonovioside A_1_ (**2**).

Position	δ_C_ mult. *^a^*	δ_H_ mult. (*J* in Hz) *^b^*	HMBC	ROESY
1	35.8 CH_2_	1.31 m		
2	26.8 CH_2_	1.98 m		
		1.78 m		
3	89.1 CH	3.19 m	C: 1 Xyl1	H-5, H-31, H1-Xyl1
4	39.3 C			
5	47.7 CH	0.91 m		H-3, H-31
6	23.1 CH_2_	1.94 m		H-19, H-30
7	120.3 CH	5.58 m		H-15, H-32
8	145.5 C			
9	47.0 CH	3.27 brd (14.7)		H-19
10	35.653 C			
11	22.4 CH_2_	1.73 m		
		1.44 m		
12	31.2 CH_2_	2.08 m	C: 11, 13, 14, 18	H-21
13	58.3 C			
14	47.5 C			
15	43.4 CH_2_	2.61 dd (7.4; 12.3)	C: 13, 14, 17, 32	H-7
		1.55 t (7.4)		
16	76.7 CH	5.85 dd (8,5; 16.6)	C: 13, OAc	H-32
17	56.8 CH	3.19 d (9.0)	C: 12, 13, 18, 20, 21	H-12, H-21, H-32
18	180.4 C			
19	23.7 CH_3_	1.10 s	C: 1, 5, 9	H-1, H-6, H-9
20	87.3 C			
21	22.4 CH_3_	1.71 s	C: 17, 20, 22	H-12, H-17
22	78.3 CH	5.71 s	C: 20, 21, 23	H-24, OAc
23	213.4 CH			
24	48.5 CH_2_	2.92 dd (6.3; 17.3)	C: 23, 25, 26, 27	
		2.73 dd (6.3; 17.3)	C: 23, 25, 26, 27	
25	23.6 CH	2.18 dd (6.3; 13.4)	C: 23, 24, 26, 27	
26	22.1 CH_3_	0.85 d (6.6)	C: 24, 27	H-25
27	22.5 CH_3_	0.89 d (6.6)	C: 24, 26	H-25
30	17.2 CH_3_	1.01 s	C: 3, 4, 5, 31	H-2, H-6, H-31
31	28.5 CH_3_	1.17 s	C: 3, 4, 5, 30	H-3, H-5, H-6, H-30, H-1 Xyl1
32	32.0 CH_3_	1.08 s	C: 8, 13, 14, 15	H-7, H-15, H-16, H-17
OCOCH_3_	169.7 C			
OCOCH_3_	21.0 CH_3_	1.86 s	C: 16, OAc	H-17, H-22

*^a^* Recorded at 125.67 MHz in C_5_D_5_N/D_2_O (4/1). *^b^* Recorded at 500.12 MHz in C_5_D_5_N/D_2_O (4/1). The original spectra of **2** are provided as Appendix A.

**Table 4 ijms-24-11128-t004:** ^13^C and ^1^H NMR chemical shifts, HMBC and ROESY correlations of aglycone moiety of djakonovioside A_2_ (**3**).

Position	δ_C_ mult. *^a^*	δ_H_ mult. (*J* in Hz) *^b^*	HMBC	ROESY
1	36.0 CH_2_	1.33 m	C: 31	H-3, H-11, H-19
2	26.8 CH_2_	1.98 m		
		1.79 m		H-19, H-30
3	89.1 CH	3.19 dd (3.6; 11.6)	C: 4, 30, 31, C: 1 Xyl1	H-1, H-5, H-31, H1-Xyl1
4	39.3 C			
5	47.5 CH	0.92 m	C: 1, 4, 30	H-3, H-31
6	23.1 CH_2_	1.92 m		H-19, H-30, H-31
7	120.2 CH	5.59 m		H-15, H-32
8	146.4 C			
9	47.3 CH	3.37 brd (14.3)		H-19
10	35.3 C			
11	22.3 CH_2_	1.73 m		
		1.41 m		H-32
12	29.7 CH_2_	1.97 m	C: 11, 13, 14, 18	H-17
13	56.9 C			
14	48.6 C			
15	43.9 CH_2_	2.42 dd (7.1; 12.9)	C: 13, 14, 17, 32	H-7, H-32
		1.83 m	C: 14, 16, 32	
16	70.2 CH	4.87 dd (6.6; 13.3)	C: 13, 14, 23 (weak)	H-32
17	50.2 CH	2.48 d (7.1)	C: 12, 13, 18, 21	H-12, H-21, H-32
18	180.3 C			
19	23.7 CH_3_	1.10 s	C: 1, 9, 10	H-1, H-2, H-9
20	82.2 C			
21	24.9 CH_3_	1.78 s	C: 17, 20, 22	H-17, H-22
22	70.9 CH	3.96 s	C: 17, 20, 21	H-21, H-24, H-27
23	96.2 C			
24	50.4 CH_2_	1.87 m	C: 22, 23, 25, 26, 27	H-22
		1.73 dd (5.7; 14.3)	C: 22, 23, 25, 26, 27	H-22, H-27
25	23.4 CH	2.15 ddd (6.1; 6.8; 13.6)	C: 23, 24, 26, 27	
26	24.5 CH_3_	0.92 d (6.5)	C: 24, 25, 27	H-25
27	24.5 CH_3_	0.89 d (6.5)	C: 24, 25, 26	H-25
30	17.3 CH_3_	1.01 s	C: 3, 4, 5, 31	H-2, H-6, H-31
31	28.6 CH_3_	1.17 s	C: 3, 4, 5, 30	H-3, H-5, H-6, H-30, H-1 Xyl1
32	33.5 CH_3_	1.05 s	C: 8, 13, 14, 15	H-7, H-11, H-12, H-15, H-16, H-17

*^a^* Recorded at 125.67 MHz in C_5_D_5_N/D_2_O (4/1). *^b^* Recorded at 500.12 MHz in C_5_D_5_N/D_2_O (4/1). The original spectra of **3** are provided as Appendix A.

**Table 5 ijms-24-11128-t005:** ^13^C and ^1^H NMR chemical shifts, HMBC and ROESY correlations of carbohydrate moiety of djaronovioside B_1_ (**4**).

Atom	δ_C_ mult. *^a, b, c^*	δ_H_ mult. (*J* in Hz) *^d^*	HMBC	ROESY
Xyl1 (1→C-3)				
1	105.3 CH	4.72 d (7.0)	C: 3; C: 5 Xyl1	H-3; H-3, 5 Xyl1
2	**82.0** CH	3.98 dd (7.0; 9.5)	C: 1 Qui2; C: 1, 3 Xyl1	H-1 Qui2; H-4 Xyl1
3	76.0 CH	4.30 t (8.9)	C: 2, 4 Xyl1	H-1, 5 Xyl1
4	*76.8* CH	4.99 dd (5.7; 8.9; 13.8)	C: 3 Xyl1	H-2 Xyl1
5	64.8 CH_2_	4.77 dd (5.7; 13.8)	C: 1, 3 Xyl1	
		3.83 m		H-1, 3 Xyl1
Qui2 (1→2Xyl1)				
1	105.2 CH	5.21 d (7.3)	C: 2 Xyl1	H-2 Xyl1; H-3, 5 Qui2
2	**83.1** CH	3.93 t (8.7)	C: 3 Qui2; C: 1 Xyl5	H-4 Qui2
3	75.8 CH	3.98 t (8.7)	C: 2, 4 Qui2	H-1, 5 Qui2
4	**85.9** CH	3.47 t (8.7)	C: 3, 5, 6 Qui2; C: 1 Xyl3	H-1 Xyl3; H-2 Qui2
5	71.7 CH	3.57 dd (6.0; 8.7)	C: 4 Qui2	H-1 Qui2
6	18.5 CH_3_	1.56 d (6.0)	C: 4, 5 Qui2	
Xyl3 (1→4Qui2)				
1	105.1 CH	4.75 d (7.7)	C: 4 Qui2	H-4 Qui2; H-3, 5 Xyl3
2	74.1 CH	3.86 t (8.9)	C: 1 Xyl3	
3	**87.0** CH	4.12 t (8.9)	C: 2, 4 Xyl3	H-1 MeGlc4; H-1 Xyl3
4	69.5 CH	3.94 t (8.9)		
5	66.6 CH_2_	4.13 m	C: 1, 3 Xyl3	
		3.59 t (10.8)	C: 1, 4 Xyl3	H-1, 3 Xyl3
MeGlc4 (1→3Xyl3)				
1	105.9 CH	5.21 d (8.3)	C: 3 Xyl3	H-3 Xyl3; H-3, 5 MeGlc4
2	75.3 CH	3.87 t (8.9)		
3	87.7 CH	3.66 t (8.9)	C: 2, 4 MeGlc4; OMe	H-1, 5 MeGlc4
4	71.1 CH	3.87 t (8.9)	C: 3, 5, 6 MeGlc4	
5	78.3 CH	3.90 m	C: 6 MeGlc4	H-1 MeGlc4
6	62.5 CH_2_	4.37 dd (1.8; 11.7)		
		4.04 dd (5.8; 11.7)	C: 5 MeGlc4	
OMe	61.5 CH_3_	3.79 s	C: 3 MeGlc4	
Xyl5 (1→2Qui2)				
1	102.6 CH	5.21 d (6.9)	C: 2 Qui2; X: 5Xyl5	H-2 Qui2
2	75.4 CH	3.96 t (8.3)	C: 3 Xyl5	
3	77.1 CH	4.05 t (8.3)	C: 2, 4 Xyl5	H-1, 5 Xyl5
4	70.8 CH	4.10 m	C: 3 Xyl5	
5	67.1 CH_2_	4.32 dd (5.0; 11.0)	C: 3, 4 Xyl5	
		3.61 m	C: 3, 4 Xyl5	H-1 Xyl5

*^a^* Recorded at 125.67 MHz in C_5_D_5_N. *^b^* Bold = interglycosidic positions. *^c^* Italic = sulfate position. *^d^* Recorded at 500.12 MHz in C_5_D_5_N. Multiplicity by 1D TOCSY. The original spectra of **4** are provided as Appendix A.

**Table 6 ijms-24-11128-t006:** ^13^C and ^1^H NMR chemical shifts, HMBC and ROESY correlations of aglycone moiety of djakonovioside B_2_ (**5**).

Position	δ_C_ mult. *^a^*	δ_H_ mult. (*J* in Hz) *^b^*	HMBC	ROESY
1	36.0 CH_2_	1.31 m		H-3, H-5, H-11, H-19
2	26.7 CH_2_	1.93 m		
		1.76 m		H-19, H-30
3	89.0 CH	3.17 dd (4.0; 11.8)	C: 4, 30, 31, C: 1 Xyl1	H-1, H-5, H-31, H1-Xyl1
4	39.3 C			
5	47.6 CH	0.91 m	C: 4, 10, 30	H-3, H-31
6	23.1 CH_2_	1.90 m		H-19, H-30, H-31
7	120.2 CH	5.60 m	C: 9	H-15, H-32
8	146.4 C			
9	47.3 CH	3.40 brd (14.0)		H-19
10	35.3 C			
11	22.3 CH_2_	1.71 m		H-1
		1.42 m		H-32
12	29.8 CH_2_	1.98 m	C: 11, 13, 18	
13	58.0 C			
14	48.5 C			
15	43.3 CH_2_	2.43 dd (7.0; 12.7)	C: 13, 14, 17, 32	H-7, H-17, H-32
		1.83 dd (7.5; 12.7)	C: 8, 14, 16, 32	
16	69.9 CH	5.03 m	C: 13, 23	H-32
17	51.0 CH	2.43 d (7.0)	C: 12, 13, 18, 21	H-15, H-21, H-32
18	180.7 C			
19	23.7 CH_3_	1.08 s	C: 1, 5, 9, 10	H-1, H-2, H-6, H-9
20	80.0 C			
21	28.7 CH_3_	1.54 s	C: 17, 20, 22	H-17, H-22
22	42.1 CH	2.39 brd (15.8)	C: 17, 20, 21, 24, 23	H-21, H-24
		1.95 brd (15.1)	C: 21, 23	H-17, H-21, H-24
23	96.7 C			
24	51.9 CH_2_	1.74 dd (2.5; 6.3)	C: 22, 23, 25, 26, 27	H-22, H-26, H-27
25	23.7 CH_2_	1.98 m	C: 23, 24, 26, 27	
26	24.4 CH_3_	0.91 d (6.5)	C: 24, 25, 27	H-24, H-25
27	24.4 CH_3_	0.87 d (6.5)	C: 24, 25, 26	H-24, H-25
30	17.3 CH_3_	1.01 s	C: 3, 4, 5, 31	H-2, H-6, H-31
31	28.6 CH_3_	1.17 s	C: 3, 4, 5, 30	H-3, H-5, H-6, H-30, H-1 Xyl1
32	33.0 CH_3_	1.06 s	C: 8, 13, 14, 15	H-7, H-11, H-12, H-16, H-17

*^a^* Recorded at 125.67 MHz in C_5_D_5_N/D_2_O (4/1). *^b^* Recorded at 500.12 MHz in C_5_D_5_N/D_2_O (4/1). The original spectra of **5** are provided as Appendix A.

**Table 7 ijms-24-11128-t007:** ^13^C and ^1^H NMR chemical shifts, HMBC and ROESY correlations of aglycone moiety of djakonovioside B_3_ (**6**).

Position	δ_C_ mult. *^a^*	δ_H_ mult. (*J* in Hz) *^b^*	HMBC	ROESY
1	35.6 CH_2_	1.32 m		H-3
2	26.7 CH_2_	1.89 m		
		1.71 m		
3	88.9 CH	3.15 dd (4.1; 11.8)	C: 4, 30, 31, C: 1 Xyl1	H-1, H-5, H-31, H1-Xyl1
4	39.3 C			
5	47.6 CH	0.85 m	C: 4, 10, 30	H-3, H-31
6	22.9 CH_2_	1.89 m		H-31
		1.80 m		H-19, H-30
7	122.4 CH	5.57 m	C: 9	H-15, H-32
8	147.7 C			
9	46.1 CH	3.03 brd (13.6)		H-19
10	35.4 C			
11	21.8 CH_2_	1.94 m		
		1.44 m		
12	20.2 CH_2_	2.49 dd (8.8; 13.0)	C: 13, 14, 18	
		2.24 dd (3.6; 13.0)	C: 13, 14, 18	H-21, H-32
13	54.8 C			
14	45.9 C			
15	44.3 CH_2_	2.10 m	C: 14, 16, 17, 32	H-7, H-17, H-32
		2.06 m	C: 14, 32	
16	80.0 CH	5.10 brs	C: 13, 14, 18	H-21, H-22
17	63.5 CH	2.81 s	C: 13, 14, 18, 20, 21, 22	H-12, H-15, H-21, H-22, H-23, H-32
18	182.8 C			
19	23.9 CH_3_	0.91 s	C: 1, 5, 9, 10	H-1, H-2, H-6, H-9, H-30
20	72.8 C			
21	26.4 CH_3_	1.58 s	C: 17, 20, 22	H-12, H-16, H-17, H-23
22	46.7 CH	1.97 m	C: 20, 21, 23, 27	H-16
		1.78 m		H-16, H-17, H-21, H-24
23	66.5 CH	4.34 m		H-21, H-27
24	48.0 CH_2_	1.61 m	C: 22, 23, 25, 26, 27	H-22, H-27
		1.26 ddd (4.7; 8.3; 13.0)	C: 22, 23, 25, 26, 27	H-22, H-26, H-27
25	24.3 CH	1.90 m	C: 24, 25, 26, 27	
26	23.2 CH_3_	0.84 d (6.5)	C: 24, 25, 27	H-24, H-25
27	22.2 CH_3_	0.88 d (6.5)	C: 24, 25, 26	H-23, H-24, H-25
30	17.2 CH_3_	0.99 s	C: 3, 4, 5, 31	H-2, H-6, H-31
31	28.6 CH_3_	1.16 s	C: 3, 4, 5, 30	H-3, H-5, H-6, H-30, H-1 Xyl1
32	34.3 CH_3_	1.45 s	C: 8, 13, 14, 15	H-7, H-12, H-15, H-17

*^a^* Recorded at 125.67 MHz in C_5_D_5_N/D_2_O (4/1). *^b^* Recorded at 500.12 MHz in C_5_D_5_N/D_2_O (4/1). The original spectra of **5** are provided as Appendix A.

**Table 8 ijms-24-11128-t008:** The cytotoxic activities of glycosides **1**–**7**, cucumarioside A_0_-1, frondoside D, okhotoside A_1_-1, and cisplatin (positive control) against human erythrocytes, HL-60, HEK293, MCF-7, T-47D, and MDA-MB-231 human cell lines.

Glycosides	ED_50_, µM, Erythrocytes	Cytotoxicity, IC_50_ µM				
		HL-60	HEK293	MCF-7	T-47D	MDA-MB-231
djakonovioside A (**1**)	2.46 ± 0.22	6.05 ± 0.17	6.36 ± 0.17	12.94 ± 0.53	11.84 ± 0.54	5.89 ± 0.11
djakonovioside A_1_ (**2**)	2.26 ± 0.25	4.44 ± 0.37	10.22 ± 1.67	26.85 ± 0.30	14.79 ± 1.73	9.64 ± 0.17
djakonovioside A_2_ (**3**)	30.86 ± 0.97	>50.0	>50.0	>50.0	>50.0	>50.0
djakonovioside B_1_ (**4**)	6.03 ± 0.47	20.06 ± 0.80	15.76 ± 0.91	22.75 ± 0.67	24.73 ± 0.20	12.43 ± 0.59
djakonovioside B_2_ (**5**)	16.34 ± 0.34	38.64 ± 1.05	25.35 ± 1.00	49.35 ± 1.12	37.68 ± 1.53	40.78 ± 0.60
djakonovioside B_3_ (**6**)	17.77 ± 0.91	49.31 ± 4.13	48.55 ± 2.15	>50.0	>50.0	>50.0
djakonovioside B_4_ (**7**)	17.07 ± 1.15	>50.0	>50.0	>50.0	>50.0	>50.0
cucumarioside A_0_-1	1.58 ± 0.56	10.99 ± 1.14	4.52 ± 0.12	12.77 ± 0.80	4.51 ± 0.93	6.04 ± 0.47
frondoside D	9.22 ± 0.10	18.08 ± 0.08	9.45 ± 0.78	23.18 ± 0.92	11.85 ± 1.10	12.03 ± 1.28
okhotoside A_1_-1	0.75 ± 0.09	2.52 ± 0.26	3.17 ± 0.70	8.34 ± 0.12	8.46 ± 0.65	2.34 ± 0.53
chitonoidoside L	1.16 ± 0.10	-	-	-	-	-
cisplatin	-	10.32 ± 1.65	151.77 ± 2.13	116.48 ± 3.15	>160.0	80.64 ± 4.12

## Data Availability

Not applicable.

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
