# Peer review of "Djakonoviosides A, A1, A2, B1–B4 — Triterpene Monosulfated Tetra- and Pentaosides from the Sea Cucumber Cucumaria djakonovi: The First Finding of a Hemiketal Fragment in the Aglycones; Activity against Human Breast Cancer Cell Lines"

_ijms, 2023, doi:10.3390/ijms241311128_

Round 1

Reviewer 1 Report

There is a need to improve several terms

Please replace as follows:

keto group with oxo group

quinovopyranosyl with 6-deoxyglucopyranosyl

acetic group with acetyl group

4-O-sodium sulfate with 4-O-sulfo ..., sodium salt

first found with found for the first time

hemiketal with hemiacetal

English language is good enough

Author Response

There is a need to improve several terms

Please replace as follows:

Note: keto group with oxo group

Reply: we have replaced.

Note: quinovopyranosyl with 6-deoxyglucopyranosyl

Reply: there is no necessity because the word “quinovose” is quite legitime and traditional name of 6-deoxyglucose.

Note: acetic group with acetyl group

Reply: we have replaced

Note: 4-O-sodium sulfate with 4-O-sulfo ..., sodium salt

Reply: Our mode of the sulfate writing seems to be simpler. Moreover, it is traditional for many dozen articles published by us and our colleagues. We should be very appreciated to the editors for keeping this tradition in the current article too.

Note: first found with found for the first time

Reply: We have replaced.

Note: hemiketal with hemiacetal

Reply: the term “hemiketal” is more certain than “hemiacetal” although the both of them are correct. Hence, we wish to conserve “hemiketal”.

All the corrections are marked in the text by yellow.

Reviewer 2 Report

In this study, Djakonoviosides -Triterpene Monosulfated Tetra- and Pentaosides from a sea cucumber Cu-cumaria djakonovi was isolated and characterized. Then their activity against some cancer cell lines were tested and their results were evaluated. 

The idea of this study and the work done is appreciated. It is a tedious work so there is novelty.

I have some minor comments:

I suggest the authors to select a much simpler title for their work. The title is almost like an abstract. The details are given in the abstract and throughout the MS. However, the complexity of the title distracts the reader.

On the other hand, introduction part is sufficient, and the reader can understand the aim of the study and can get the idea of what have been performed.

For Table 8, I recommend the authors to apply statistics to address the difference between glycosides for each test.

I also find the discussion and conclusion parts sufficient. 

Author Response

In this study, Djakonoviosides -Triterpene Monosulfated Tetra- and Pentaosides from a sea cucumber Cu-cumaria djakonovi was isolated and characterized. Then their activity against some cancer cell lines were tested and their results were evaluated. 

The idea of this study and the work done is appreciated. It is a tedious work so there is novelty.

I have some minor comments:

Note: I suggest the authors to select a much simpler title for their work. The title is almost like an abstract. The details are given in the abstract and throughout the MS. However, the complexity of the title distracts the reader.

On the other hand, introduction part is sufficient, and the reader can understand the aim of the study and can get the idea of what have been performed.

Reply: the title is compressed.

Note: For Table 8, I recommend the authors to apply statistics to address the difference between glycosides for each test.

I also find the discussion and conclusion parts sufficient.

Reply: There is no sense just now in such procedure because the number of substances seems to be too low and the SAR trends may be found “manually” without the use special statistic program. Moreover, all the found trends are in good correlations with earlier known. However, we intend to carry out such investigation after completing the chemical study of the glycosides from Cucumaria djakonovi when we shall obtain more substances.

All the corrections are marked with yellow.